# SELF-SUPERVISED MONOCULAR DEPTH ESTIMATION ROBUST TO REFLECTIVE SURFACE LEVERAGED BY TRIPLET MINING

**Wonhyeok Choi**[1,*]**, Kyumin Hwang**[1,*]**, Wei Peng**[2]**, Minwoo Choi**[1]**, Sunghoon Im**[1,†]
Electrical Engineering and Computer Science[1], Psychiatry and Behavioral Sciences[2]
Daegu Gyeongbuk Institute of Science and Technology[1], Stanford University[2]
South Korea[1], USA[2]
{smu06117,kyumin,subminu,sunghoonim}@dgist.ac.kr[1], wepeng@stanford.edu[2]

## ABSTRACT

Self-supervised monocular depth estimation (SSMDE) aims to predict the dense depth map of a monocular image, by learning depth from RGB image sequences, eliminating the need for ground-truth depth labels. Although this approach simplifies data acquisition compared to supervised methods, it struggles with reflective surfaces, as they violate the assumptions of Lambertian reflectance, leading to inaccurate training on such surfaces. To tackle this problem, we propose a novel training strategy for an SSMDE by leveraging triplet mining to pinpoint reflective regions at the pixel level, guided by the camera geometry between different viewpoints. The proposed reflection-aware triplet mining loss specifically penalizes the inappropriate photometric error minimization on the localized reflective regions while preserving depth accuracy in non-reflective areas. We also incorporate a reflection-aware knowledge distillation method that enables a student model to selectively learn the pixel-level knowledge from reflective and non-reflective regions. This results in robust depth estimation across areas. Evaluation results on multiple datasets demonstrate that our method effectively enhances depth quality on reflective surfaces and outperforms state-of-the-art SSMDE baselines.

## 1 INTRODUCTION

Self-supervised monocular depth estimation (SSMDE) (Godard et al., 2019) is a task that learns depth solely from a continuous RGB image sequence without needing corresponding ground-truth depth maps for each frame in a video. This approach significantly simplifies data acquisition compared to traditional supervised methods (Fu et al., 2018; Lee et al., 2019; Bhat et al., 2021), which often involve high costs for annotation. As such, many SSMDE studies (Godard et al., 2019; Zhou et al., 2017; Garg et al., 2016; Guizilini et al., 2020) have explored its viability as a mainstay for applications such as autonomous driving, highlighting its potential in outdoor environments.

Despite its advantages, SSMDE approaches typically challenge in accurate depth estimation on non-Lambertian surfaces such as mirrors, transparent objects, and specular surfaces. This difficulty primarily arises from the assumption of Lambertian reflectance (Basri & Jacobs, 2003) embedded in most SSMDE methods. As illustrated in Figure 1, these non-Lambertian surfaces violate the photometric constancy principle, which posits that the color and brightness of a point should appear constant across different images (Godard et al., 2017). This violation leads to incorrect depth training, particularly on non-Lambertian surfaces. Consequently, this issue manifests in a phenomenon known as the "black-hole effect" (Shi et al., 2023), where the model erroneously predicts depths that are greater than the actual surface depth in areas with specular reflections. This effect is a prevalent challenge across various reflective surfaces, significantly impacting the performance and reliability of SSMDE systems.

---

*These authors contributed equally.
†S. Im is the corresponding author.

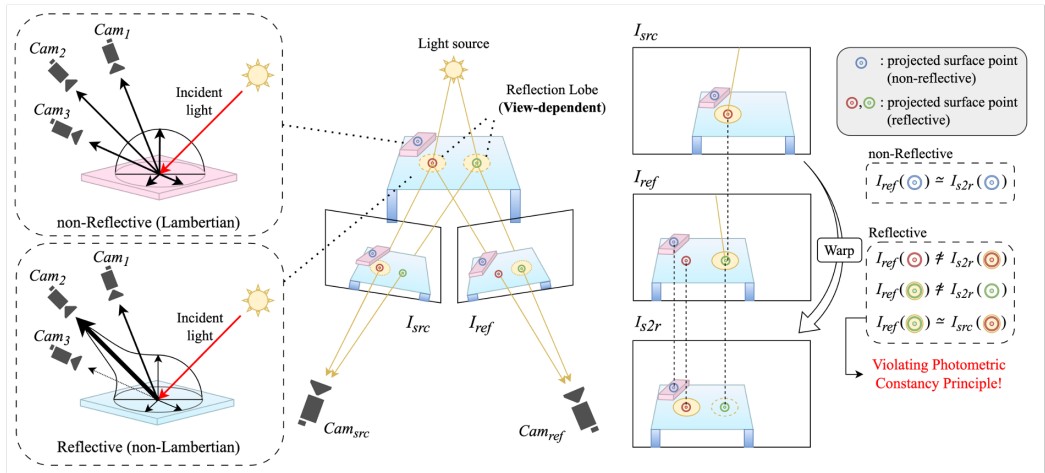

Figure 1: Photometric constancy violation on reflective surfaces. The projected non-reflective surface point (denoted as ⊙) satisfies the photometric constancy so the model can obtain the accurate depth by photometric error minimization. On the other hand, projected reflective surface point (denoted as ⊙,⊙) violates the photometric constancy, resulting in wrong disparity by photometric error minimization. This figure depicts a scenario where the relative positions of the cameras shift horizontally, akin to rectified stereo, to simplify the illustration.

Recent advancements (Costanzino et al., 2023; Shi et al., 2023) attempt to tackle these challenges by utilizing training strategies that involve generating pseudo-labels through inpainting (Costanzino et al., 2023) or reconstructing 3D meshes (Shi et al., 2023). However, these methods still rely on extra labels such as segmentation mask annotations (Costanzino et al., 2023) or use auxiliary methods that have excessive computational costs such as ensemble-based uncertainty algorithms (Shi et al., 2023), TSDF-fusion (Newcombe et al., 2011) and mesh rendering (Matl, 2019).

To address these issues, we propose a novel training strategy called "*reflection-aware triplet mining*" that enhances the performance of SSMDEs by leveraging the triplet mining (Schroff et al., 2015). The underlying principle of our approach is that reflective areas, such as mirror light sources or objects, exhibit disparities corresponding to the reflected object rather than the actual surface as illustrated in Figure 1. While non-reflective areas exhibit photometric error due to the difference in camera views from two different perspectives (*e.g.*, source and reference views), reflective areas have abnormally low photometric error between the two views due to the low disparity of reflected objects. Accordingly, our approach treats views from the same camera coordinates as positive pairs and those from different coordinates as negative pairs, as illustrated in Figure 2. Our method aims to minimize the conventional photometric error between positive pairs while maximizing it between negative pairs. This approach effectively neutralizes the impact of contaminated gradients in reflective regions, thereby significantly improving accuracy on these regions.

Moreover, we introduce a "*reflection-aware knowledge distillation*" approach to keep the high-frequency details in the predicted depth for non-reflective regions inspired by previous works (Shi et al., 2023). In this method, the student network is trained by distilling knowledge from two distinct SSMDE networks. The first utilizes the proposed triplet loss, providing robustness against reflective areas, while the second employs solely photometric minimization loss, adept at preserving high-frequency details that contribute to the perceptual quality and visual fidelity of the depth map. This hybrid training strategy effectively combines the strengths of both training methods, creating a more versatile and effective depth estimation framework. By leveraging the unique capabilities of each model, the student network can achieve a more comprehensive understanding of depth across various surface conditions.

Our method is broadly applicable to general SSMDE frameworks that rely on photometric error minimization. We validate our method on three well-known SSMDE networks (Godard et al., 2019; Lyu et al., 2021; Zhao et al., 2022) across three public datasets (Dai et al., 2017; Shotton et al.,

2013; Ramirez et al., 2023) featuring reflective objects and surfaces. The results demonstrate that our method significantly improves depth prediction accuracy on reflective surfaces while preserving performance on non-reflective surfaces. Our main contributions are fourfold as follows:

1. We propose a new *reflection-aware triplet mining loss* that significantly enhances the accuracy on reflective surfaces and can be easily integrated into general SSMDE frameworks.

2. We introduce *reflection-aware knowledge distillation* to improve the overall accuracy on reflective surfaces while preserving high-frequency details on non-reflective surfaces.

3. To the best of our knowledge, our strategy represents the first end-to-end method specifically designed to enhance the performances of SSMDE on reflective surfaces.

4. The proposed method outperforms the existing self-supervised training method and shows comparable results against 3D information distillation methods on various indoor depth benchmarks.

## 2 RELATED WORK

### 2.1 SELF-SUPERVISED MONOCULAR DEPTH ESTIMATION

Self-supervised Monocular Depth Estimation (SSMDE) is a task that estimates a depth map from a single image without a ground truth depth map. This approach significantly simplifies the process of data acquisition, making it scalable for a wide variety of datasets. SfMLearner (Zhou et al., 2017) introduces a pioneering framework for self-supervised depth map estimation, which simultaneously learns depth maps for the input image and pose parameters from sequential views. Monodepth2 (Godard et al., 2019) proposes a masking scheme and minimum reprojection loss to filter out the regions that violate photometric constancy, such as moving objects and occluded regions. Subsequent methods (Zhou et al., 2021; Guizilini et al., 2020; Lyu et al., 2021) have been refined, effectively integrating features of different resolutions based on established constraints. With the introduction of ViT (Dosovitskiy et al., 2020), the field of SSMDE has begun to incorporate transformer backbones. Monoformer (Bae et al., 2023b) and MonoViT (Zhao et al., 2022) have emerged, utilizing hybrid networks of CNN and transformers to adeptly merge local and global features.

### 2.2 GENERALIZATION OF MONOCULAR DEPTH ESTIMATION

Recent research has expanded to consider factors such as the impact of weather variations (Saunders et al., 2023; Gasperini et al., 2023), the differences in inference capabilities between CNNs and Transformers (Bae et al., 2023a), the robustness of SSMDEs against various types of data corruption (Kong et al., 2024), and methods for accurately modeling transparent and mirrored surfaces, which are typically non-Lambertian (Costanzino et al., 2023). In addition, the 3D Distillation (Shi et al., 2023) addresses a critical flaw in traditional SSMDEs: the photometric constancy principle used in applying photometric consistency loss may not hold for non-Lambertian surfaces encountered in real-world scenarios, resulting in SSMDE models producing unreliable and low-quality depth estimates for reflective surfaces. To counter this problem, 3D Distillation leverages the 3D mesh rendering function along with ensemble uncertainty to localize the reflective surfaces and refine the inaccurate depth on these regions.

### 2.3 DEEP METRIC LEARNING

Deep metric learning (Chen et al., 2020; Chen & He, 2021; Khosla et al., 2020) seeks to develop an effective distance measure between data points. These methods strive to minimize the distance between samples of the same class while maximizing it between samples from different classes. While traditionally focused on classification tasks, where positive and negative pairs are defined based on class similarity, recent studies (Spurr et al., 2021; Wang et al., 2022; Zha et al., 2024) have expanded the application of deep metric learning to regression contexts. Particularly in the context of depth estimation, deep metric learning has demonstrated versatility beyond simple augmentation-based consistency. It has been applied to enhance accuracy through contrasting depth distributions (Fan et al., 2023; Choi et al., 2024) and addressing issues such as edge fattening (Chen et al., 2023). In this paper, we utilize the triplet mining scheme, initially popularized by Schroff et al. (2015), to enhance recognition accuracy, specifically focusing on improving performance on reflective surfaces.

## 3 METHOD

Our method aims to enhance depth prediction accuracy on reflective surfaces by strategically penalizing the inappropriate photometric error minimization between the view-synthesized image and the reference image. In Section 3.1, we discuss the photometric constancy principle, which posits that correctly minimizing photometric error is crucial for accurately determining depth (Section 3.1.1). We also provide an overview of the standard training strategies employed in SSMDE frameworks (Section 3.1.2). In Section 3.2, we detail the three components of our training strategy: reflective region localization (Section 3.2.1), reflection-aware triplet mining loss (Section 3.2.2), and reflection-aware knowledge distillation (Section 3.2.3).

### 3.1 PRELIMINARY

#### 3.1.1 PHOTOMETRIC CONSTANCY PRINCIPLE

The photometric constancy principle is foundational in SSMDE frameworks, positing that surfaces exhibit uniform reflectance (*i.e.*, Lambertian reflectance) from all viewing angles. A surface adheres to this principle if its color and luminance observed through a camera remain constant, regardless of the camera's viewing angle. By leveraging this property, depth and pose can be accurately estimated by minimizing the photometric error between the view-synthesized image and the reference image, as described in Equation 3. However, real-world scenes rarely adhere strictly to this principle. Non-Lambertian surfaces, such as specular reflections from light sources or mirrored objects, are prevalent, leading to violations of photometric constancy. These deviations result in significant errors when attempting to minimize photometric error, thus compromising the effectiveness of depth estimation methods based on these assumptions.

#### 3.1.2 TRAINING STRATEGY OF GENERAL SSMDE FRAMEWORK

The objective of SSMDE is to predict a per-pixel basis depth map $\mathbf{D}_{ref}$ of a reference image $\mathbf{I}_{ref}$ given the reference image itself, a source image $\mathbf{I}_{src}$ (or source images) and their camera intrinsics $\boldsymbol{K}$. The framework consists of a depth network $\mathcal{F}_\theta(\cdot)$, and a pose network $\mathcal{G}_\phi(\cdot, \cdot)$ to respectively estimate the depth of the reference image $\mathbf{D}_{ref}$, and the relative pose $[\boldsymbol{R}|\boldsymbol{t}]_{r2s}$ as follows:

$$\mathbf{D}_{ref} = \mathcal{F}_\theta(\mathbf{I}_{ref}), \qquad \mathcal{F}_\theta : \mathbb{R}^{3\times h\times w} \quad \rightarrow \mathbb{R}^{1\times h\times w}, \tag{1}$$

$$[\boldsymbol{R}|\boldsymbol{t}]_{r2s} = \mathcal{G}_\phi(\mathbf{I}_{src}, \mathbf{I}_{ref}), \quad \mathcal{G}_\phi : \mathbb{R}^{2\times 3\times h\times w} \rightarrow \mathbb{R}^{3\times 4}, \tag{2}$$

where $(h, w)$ represent the spatial resolution of $\mathbf{I}_{ref}$. Using the obtained relative pose $[\boldsymbol{R}|\boldsymbol{t}]_{r2s}$, and depth map $\mathbf{D}_{ref}$, the source image $\mathbf{I}_{src}$ is warped into the reference coordinates, generating the view-synthesized image $\mathbf{I}_{s2r}$ as follows:

$$(\mathbf{I}_{s2r})_{:,u,v} = \mathbf{I}_{src}(\langle \boldsymbol{K}[\boldsymbol{R}|\boldsymbol{t}]_{r2s}(\mathbf{D}_{ref})_{:,u,v}\boldsymbol{K}^{-1}[u, v, 1]^T\rangle), \tag{3}$$

where $(u, v)$ represent the image coordinates and $\langle\cdot\rangle$ is the projection function that maps homogeneous coordinates to image coordinates. By the photometric constancy principle detailed in Section 3.1.1, the synthesized image $\mathbf{I}_{s2r}$ should exhibit the same colors and luminances as the reference image on a pixel-by-pixel basis. Consequently, the model can determine the accurate depth and pose by minimizing the photometric errors, $\mathcal{P}(\cdot, \cdot)$, between $\mathbf{I}_{s2r}$ and $\mathbf{I}_{ref}$ as follows:

$$\mathcal{P}(\mathbf{I}_{s2r}, \mathbf{I}_{ref}) = \mathbf{M} \odot \left( \alpha_1 \frac{1 - \mathcal{S}(\mathbf{I}_{ref}, \mathbf{I}_{s2r})}{2} + \alpha_2 ||\mathbf{I}_{ref} - \mathbf{I}_{s2r}||_1 \right),$$
$$\mathcal{P} : \mathbb{R}^{2\times 3\times h\times w} \rightarrow \mathbb{R}^{1\times h\times w}, \quad \mathcal{S} : \mathbb{R}^{2\times 3\times h\times w} \rightarrow \mathbb{R}^{1\times h\times w}, \tag{4}$$

where $\mathcal{S}(\cdot, \cdot)$ is the mixture of structural similarity index (Wang et al., 2004), and $\mathbf{M}$ is the principled mask (Godard et al., 2019) to prevent the backpropagation of corrupted gradients, caused by anomalies like moving objects in the scene. The weights $\alpha_1$ and $\alpha_2$ serve as balance terms between two losses, and $\odot$ represents the element-wise multiplication operator. However, if the surface does not conform to the principle of photometric constancy, minimizing photometric errors on such reflective surfaces can lead to significant inaccuracies in the estimated depth.

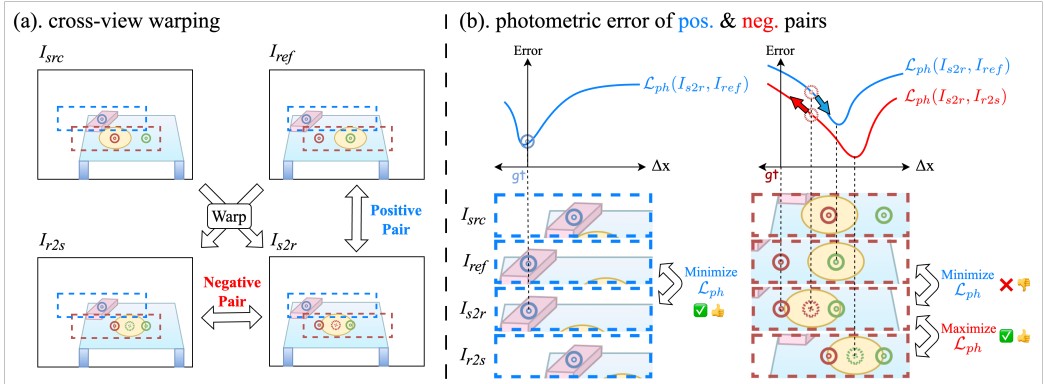

Figure 2: The effect of the proposed method on reflective/non-reflective surfaces. (⊚/⊙,⊙) imply the projected non-reflective/reflective surface points, respectively, and (❀, ❀) denotes the location of reflection lobe in view-synthesized image coordinate. Our proposed method cancels out the wrong photometric error minimization in reflection areas by contrasting the negative pair samples.

## 3.2 METHODOLOGY

### 3.2.1 REFLECTIVE REGION LOCALIZATION

The photometric error, as calculated through Equation 4 between $I_{s2r}$ and $I_{ref}$ tends to be smaller in non-reflective regions. This is because these areas reflect consistent colors and luminances irrespective of the viewing direction, adhering to the photometric constancy principle. On the other hand, reflective regions, which violate the photometric constancy principle, exhibit abnormally low disparities due to the additional distance from the reflected light source. Consequently, as illustrated in Figure 2, reflection lobes from different images appear relatively closer in image coordinates, resulting in reduced photometric errors in the RGB images of two different viewpoints.

This characteristic is crucial for isolating reflective regions within the spatial dimension of the image. To capitalize on this, we first generate cross-view synthesized images $I_{s2r}, I_{r2s}$ in a manner similar to the process outlined in Equation 3:

$$(\mathbf{I}_{s2r})_{:,u,v} = \mathbf{I}_{src}(\langle \boldsymbol{K}[\boldsymbol{R}|\boldsymbol{t}]_{r2s}(\mathbf{D}_{ref})_{:,u,v}\boldsymbol{K}^{-1}[u,v,1]^T \rangle), \tag{5}$$

$$(\mathbf{I}_{r2s})_{:,u,v} = \mathbf{I}_{ref}(\langle \boldsymbol{K}[\boldsymbol{R}|\boldsymbol{t}]_{s2r}(\mathbf{D}_{src})_{:,u,v}\boldsymbol{K}^{-1}[u,v,1]^T \rangle), \tag{6}$$

where the relative pose $[\boldsymbol{R}|\boldsymbol{t}]_{s2r}$ can be obtained by computing the inverse of the predicted pose $[\boldsymbol{R}|\boldsymbol{t}]_{r2s}$, and $\mathbf{D}_{src}$ is predicted depth from $\mathbf{I}_{src}$, following a procedure similar to Equation 1. Utilizing these synthesized cross-view images, we compute two photometric errors to measure discrepancies between image pairs as follows:

$$\mathbf{E}^+ = \mathcal{P}(\mathbf{I}_{s2r}, \mathbf{I}_{ref}), \quad \mathbf{E}^- = \mathcal{P}(\mathbf{I}_{s2r}, \mathbf{I}_{r2s}). \tag{7}$$

The first error, $\mathbf{E}^+$, quantifies discrepancies between images taken from the same viewpoint $(\mathbf{I}_{s2r}, \mathbf{I}_{ref})$. This error is minimized when depth and pose are accurately estimated on non-reflective surfaces. Conversely, the second error, $\mathbf{E}^-$, measures discrepancies between images from different viewpoints $(\mathbf{I}_{s2r}, \mathbf{I}_{r2s})$. In general, the expected photometric error for $\mathbf{E}^-$ should be substantial due to the different camera coordinate systems. However, on reflective surfaces, the variations in light reflection may result in a reduced photometric error. Based on these observations, we identify pixel-level reflective regions $\mathbf{M}_r \in \mathbb{R}^{1 \times h \times w}$ as follows:

$$(\mathbf{M}_r)_{:,u,v} = \begin{cases} 1, & \text{if } (\mathbf{E}^-)_{:,u,v} - (\mathbf{E}^+)_{:,u,v} \leq \delta, \\ 0, & \text{else}, \end{cases} \tag{8}$$

where $\delta$ is a certain margin that represents the minimum significant photometric difference required to distinguish between the two surface types, where a value of 1 corresponds to a reflective pixel and 0 to a non-reflective pixel, respectively. This method effectively utilizes discrepancies in photometric errors to distinguish between reflective and non-reflective surfaces, providing a precise mapping of surface properties within the image. (refer to Section D in the supplementary materials.)

### 3.2.2 REFLECTION-AWARE TRIPLET MINING LOSS

We introduce the reflection-aware triplet mining loss, $\mathcal{L}_{tri}$, which addresses the limitations of using $\mathbf{E}^+$ alone in environments where reflections disrupt depth accuracy. In reflective regions, simply minimizing $\mathbf{E}^+$ does not effectively discern between real and reflected disparities. To counteract this, we assert that $\mathbf{E}^-$ should be significantly greater than $\mathbf{E}^+$. This approach is inspired by triplet mining techniques that aim to minimize the distance within positive pairs and maximize it within negative pairs, enhancing the model's ability to distinguish between reflective and non-reflective surfaces. To implement this, we formulate the reflection-aware triplet mining loss as follows:

$$\mathcal{L}_{tri}(\mathbf{I}_{ref}, \mathbf{I}_{s2r}, \mathbf{I}_{r2s}) = \mathbf{M}_r \odot (\mathbf{E}^+ - \mathbf{E}^- + \delta)_{hinge} + (1 - \mathbf{M}_r) \odot \mathbf{E}^+, \qquad (9)$$

where $(\cdot)_{hinge}$ is the hinge loss function described in Hearst et al. (1998). In this configuration, the reflection-aware triplet mining loss is applied specifically to regions identified as reflective. For non-reflective regions, where reflections do not disrupt photometric assessments, we apply the photometric loss $\mathbf{E}^+$ as it reliably reflects photometric consistency. This differentiation allows the model to address the unique challenges presented by each type of region effectively.

As illustrated in Figure 2, this strategy involves not only penalizing the minimization of $\mathbf{E}^+$ but also actively maximizing $\mathbf{E}^-$. This method effectively counteracts the contaminated gradients typically found in reflective regions. By adjusting the balance between $\mathbf{E}^+$ and $\mathbf{E}^-$ based on the presence of reflective surfaces, our method not only improves depth estimation in complex scenarios but also ensures robust performance against the challenges posed by reflective surfaces. This comprehensive approach results in a model that accurately reflects the true topography of both reflective and non-reflective environments.

### 3.2.3 REFLECTION-AWARE KNOWLEDGE DISTILLATION

The proposed end-to-end training scheme described in Section 3.2.2 effectively handles the depth estimation on both reflective and non-reflective surfaces. To further refine depth estimation quality, we introduce a reflection-aware knowledge distillation strategy inspired by the fusion techniques discussed in Shi et al. (2023), aimed at retaining high-frequency details in depth prediction.

Our approach begins by training two separate SSMDE networks. The first is trained using our reflection-aware triplet mining loss, $\mathcal{L}_{tri}$, as defined in Equation 9, and the second employs the conventional photometric loss, $\mathbf{E}^+$. From these models, we generate two types of depth maps: $\mathbf{D}_{tri}$, derived from the reflection-aware model, and $\mathbf{D}_{ori}$, obtained from the model trained with conventional photometric loss. We then merge these depth maps into a single pseudo depth map $\mathbf{D}_{pse}$ utilizing a reflective region mask $\mathbf{M}_r$. This mask facilitates the adaptive fusing of depth information from both teacher models based on the presence of reflective properties in the image. The pseudo depth map generation and distillation process is detailed in the following equation:

$$\mathcal{L}_{rkd}(\hat{\mathbf{D}}, \mathbf{D}_{pse}) = |\log \hat{\mathbf{D}} - \log \mathbf{D}_{pse}|, \text{ where } \mathbf{D}_{pse} = \mathbf{M}_r \odot \mathbf{D}_{tri} + (1 - \mathbf{M}_r) \odot \mathbf{D}_{ori}, \quad (10)$$

where $\hat{\mathbf{D}}$ represents the depth predicted by the student model. It is important to note that the student model and the two teacher models share the same network architecture as the general SSMDEs. This structured training approach not only addresses the specific challenges posed by reflective surfaces but also ensures that the high-frequency detail is not lost, thus achieving a balanced and accurate depth prediction across different surface types.

## 4 EXPERIMENTS

**Datasets.** *ScanNet (v2)* (Dai et al., 2017) is a comprehensive indoor RGB-D video dataset comprising 2.7 million images across 1,216 interior scene sequences. Traditionally, this dataset has been pivotal for evaluating multi-view stereo (Im et al., 2019; Bae et al., 2022) and scene reconstruction applications (Murez et al., 2020; Zhou et al., 2024). *KITTI* (Geiger et al., 2013) captures autonomous driving information from outdoor scenes via cameras and lidar sensors and is the most representative and long-standing benchmark for depth estimation research. *NYU-v2* (Silberman et al., 2012) serves as one of the most established and widely used benchmarks for depth estimation, as it provides a diverse set of indoor scenes with varying lighting conditions and a wide variety of objects, captured

Table 1: Main results on the ScanNet-Reflection Test and Validation sets.

| | Backbone | Training Scheme | Method | Abs Rel ↓ | Sq Rel ↓ | RMSE ↓ | RMSE log ↓ | $\delta < 1.25$ ↑ | $\delta < 1.25^2$ ↑ | $\delta < 1.25^3$ ↑ |
|---|---|---|---|---|---|---|---|---|---|---|
| **ScanNet-Reflection Test** | Monodepth2 | End-to-End | Self-Supervised | 0.181 | 0.160 | 0.521 | 0.221 | 0.758 | 0.932 | 0.976 |
| | | | *Ours* | **0.157** | **0.096** | **0.468** | **0.201** | **0.762** | **0.949** | **0.988** |
| | | Multi-Stage | Self-Teaching | 0.179 | 0.146 | 0.502 | 0.218 | 0.750 | 0.938 | 0.980 |
| | | | 3D Distillation | 0.156 | 0.096 | 0.459 | 0.195 | 0.766 | 0.945 | 0.988 |
| | | | *Ours*† | **0.150** | **0.087** | **0.446** | **0.192** | **0.777** | **0.955** | **0.990** |
| | HRDepth | End-to-End | Self-Supervised | 0.182 | 0.168 | 0.530 | 0.225 | 0.749 | 0.937 | 0.979 |
| | | | *Ours* | **0.157** | **0.098** | **0.470** | **0.201** | **0.763** | **0.952** | **0.989** |
| | | Multi-Stage | Self-Teaching | 0.175 | 0.145 | 0.492 | 0.215 | 0.757 | 0.936 | 0.982 |
| | | | 3D Distillation | 0.152 | **0.089** | 0.451 | **0.190** | 0.771 | **0.956** | **0.990** |
| | | | *Ours*† | **0.150** | 0.092 | **0.434** | 0.192 | **0.780** | 0.950 | 0.988 |
| | MonoViT | End-to-End | Self-Supervised | 0.154 | 0.129 | 0.458 | 0.197 | 0.822 | 0.955 | 0.979 |
| | | | *Ours* | **0.136** | **0.087** | **0.414** | **0.178** | **0.831** | **0.967** | **0.988** |
| | | Multi-Stage | Self-Teaching | 0.151 | 0.130 | 0.439 | 0.191 | 0.837 | 0.950 | 0.978 |
| | | | 3D Distillation | 0.127 | **0.069** | **0.379** | **0.162** | 0.846 | 0.961 | **0.992** |
| | | | *Ours*† | **0.126** | 0.074 | 0.395 | 0.167 | **0.854** | **0.969** | 0.990 |
| **ScanNet-Reflection Validation** | Monodepth2 | End-to-End | Self-Supervised | 0.206 | 0.227 | 0.584 | 0.246 | 0.750 | 0.912 | 0.961 |
| | | | *Ours* | **0.166** | **0.125** | **0.492** | **0.209** | **0.763** | **0.934** | **0.981** |
| | | Multi-Stage | Self-Teaching | 0.192 | 0.188 | 0.548 | 0.233 | 0.764 | 0.920 | 0.967 |
| | | | 3D Distillation | 0.156 | **0.093** | **0.442** | **0.191** | 0.786 | 0.943 | **0.987** |
| | | | *Ours*† | **0.151** | 0.105 | 0.454 | 0.193 | **0.796** | **0.944** | 0.985 |
| | HRDepth | End-to-End | Self-Supervised | 0.213 | 0.244 | 0.605 | 0.255 | 0.741 | 0.906 | 0.961 |
| | | | *Ours* | **0.167** | **0.127** | **0.496** | **0.210** | **0.770** | **0.937** | **0.982** |
| | | Multi-Stage | Self-Teaching | 0.202 | 0.208 | 0.565 | 0.243 | 0.756 | 0.914 | 0.964 |
| | | | 3D Distillation | 0.153 | **0.090** | **0.430** | **0.188** | 0.789 | 0.948 | **0.989** |
| | | | *Ours*† | **0.151** | 0.104 | 0.450 | 0.192 | **0.800** | **0.949** | 0.987 |
| | MonoViT | End-to-End | Self-Supervised | 0.179 | 0.206 | 0.557 | 0.227 | 0.819 | 0.930 | 0.963 |
| | | | *Ours* | **0.139** | **0.107** | **0.452** | **0.183** | **0.836** | **0.954** | **0.984** |
| | | Multi-Stage | Self-Teaching | 0.176 | 0.195 | 0.537 | 0.224 | 0.823 | 0.930 | 0.963 |
| | | | 3D Distillation | **0.126** | **0.068** | **0.367** | **0.159** | 0.851 | **0.965** | **0.991** |
| | | | *Ours*† | 0.130 | 0.091 | 0.420 | 0.173 | **0.851** | 0.960 | 0.987 |

using a Kinect sensor. ***7-Scenes*** (Shotton et al., 2013) is a challenging RGB-D dataset captured in indoor scenes with a similar distribution to *ScanNet* but dominated by non-reflective surfaces. We follow the evaluation protocol of Long et al. (2021); Bae et al. (2022) to demonstrate the cross-dataset generalization performances. ***Booster*** (Ramirez et al., 2023) includes a variety of non-Lambertian objects within indoor settings, such as transparent basins, mirrors, and specular surfaces. Following the Costanzino et al. (2023), we use the training split as our test set, which showcases our method's adaptability to more complex scenes.

**Training scenario.** In the work of 3D distillation (Shi et al., 2023), the ScanNet dataset has been further segmented into ScanNet-Reflection and ScanNet-NoReflection subsets based on the presence of reflective objects within the scenes. This subdivision results in a ScanNet-Reflection dataset consisting of 45,539 training, 439 validation, and 121 testing samples. Additionally, a ScanNet-NoReflection validation set comprising 1,012 samples evaluates the model's generalization when trained in reflective environments. Aligning with these methodologies, the training process leverages the ScanNet-Reflection train set to simulate real-world scenarios involving reflective surfaces. For the KITTI and NYU-v2 experimental setups, we follow the training protocol of Godard et al. (2019), incorporating our reflection-aware triplet loss and distillation training procedure. All experiments conducted on KITTI and NYU-v2 utilized a ResNet-18 pose network instead of a GT pose.

**Evaluation.** For quantitative evaluations, we employ standard metrics from the depth estimation literature (Eigen et al., 2014; Geiger et al., 2012). We differentiate our training approaches into end-to-end and multi-stage (distillation) strategies to effectively assess the models. The model trained solely using reflection-aware triplet mining loss $\mathcal{L}_{tri}$, referred to as "*Ours*", and another utilizing the proposed distillation method $\mathcal{L}_{rkd}$, referred to as "*Ours*†", are evaluated under their respective conditions. We compare these against both end-to-end and multi-stage baselines across three sets: ScanNet-Reflection {Test, Validation} sets, and ScanNet-NoReflection Validation set. For the KITTI (Geiger et al., 2013) and NYU-v2 (Silberman et al., 2012) datasets, we follow the standard evaluation protocol of Godard et al. (2019). To underline the cross-dataset generalizability of our methods, we also perform zero-shot evaluations on the 7-Scenes and Booster.

Table 2: Main results on the ScanNet-NoReflection Validation set.

| Backbone | Training Scheme | Method | Abs Rel ↓ | Sq Rel ↓ | RMSE ↓ | RMSE log ↓ | $\delta < 1.25$ ↑ | $\delta < 1.25^2$ ↑ | $\delta < 1.25^3$ ↑ |
|---|---|---|---|---|---|---|---|---|---|
| Monodepth2 | End-to-End | Self-Supervised | 0.169 | 0.100 | **0.395** | **0.206** | **0.759** | **0.932** | 0.979 |
| | | *Ours* | **0.168** | **0.095** | 0.395 | 0.208 | 0.751 | 0.931 | **0.980** |
| | Multi-Stage | Self-Teaching | 0.161 | 0.090 | 0.375 | 0.196 | 0.777 | 0.939 | 0.981 |
| | | 3D Distillation | 0.159 | 0.087 | **0.373** | 0.195 | **0.779** | 0.941 | **0.983** |
| | | *Ours*$^\dagger$ | **0.157** | **0.085** | 0.373 | **0.195** | 0.776 | **0.942** | **0.983** |
| HRDepth | End-to-End | Self-Supervised | 0.169 | 0.102 | **0.388** | **0.202** | **0.766** | **0.933** | **0.980** |
| | | *Ours* | **0.167** | **0.096** | 0.389 | 0.204 | 0.764 | **0.933** | 0.979 |
| | Multi-Stage | Self-Teaching | 0.160 | 0.089 | 0.367 | 0.192 | 0.784 | 0.941 | 0.982 |
| | | 3D Distillation | 0.158 | **0.086** | **0.365** | **0.190** | **0.786** | **0.942** | **0.983** |
| | | *Ours*$^\dagger$ | **0.157** | **0.086** | 0.366 | 0.192 | 0.784 | **0.942** | **0.983** |
| MonoViT | End-to-End | Self-Supervised | **0.140** | 0.074 | **0.333** | **0.171** | **0.829** | **0.952** | 0.984 |
| | | *Ours* | 0.141 | **0.072** | 0.338 | 0.174 | 0.823 | **0.952** | **0.987** |
| | Multi-Stage | Self-Teaching | 0.134 | 0.068 | 0.317 | 0.164 | **0.840** | 0.956 | **0.987** |
| | | 3D Distillation | **0.133** | **0.065** | **0.311** | **0.162** | 0.838 | 0.956 | **0.987** |
| | | *Ours*$^\dagger$ | **0.133** | 0.066 | 0.320 | 0.166 | 0.837 | **0.957** | **0.987** |

**Implementation details.** Our experiments incorporate three leading architectures in SSMDE: ***Monodepth2*** (Godard et al., 2019), ***HRDepth*** (Lyu et al., 2021), and ***MonoViT*** (Zhao et al., 2022), which have demonstrated exceptional performance in previous studies. Each backbone is trained by different training schemes, including Self-Supervised (Godard et al., 2019), Self-Teaching (Poggi et al., 2020), and 3D Distillation (Shi et al., 2023), to compare with our method. To align closely with 3D Distillation, all training particulars follow their documented conditions, with adaptations only in our proposed training strategy. Specifically, the models are trained using the reflection triplet split introduced in 3D Distillation. To finely tune the margin $\delta$ across positive and negative pairs, it is adaptively selected based on the difference between the first quartile (Q1) of $\mathbf{E}^+$ and the third quartile (Q3) of $\mathbf{E}^-$.

## 4.1 EVALUATION ON REFLECTION DATASETS

**ScanNet-Reflection dataset.** To demonstrate the effectiveness of the proposed method on reflective surfaces, we conduct a quantitative evaluation using the ScanNet-Reflection dataset. The evaluations are divided into end-to-end and multi-stage methodologies. As depicted in Table 1, *Ours*, categorized under end-to-end training schemes, significantly outperforms self-supervised methods across all backbones, achieving an *Abs Rel* average increase of 12.90% in the test split and 21.12% in the validation split. Moreover, it is noteworthy that *Ours* shows a significant performance boost, with an average improvement of 10.75% over Self-Teaching across all metrics in both the test and validation splits, with only two exceptions in 42 metrics ($\delta < 1.25$ of Monodepth2 and MonoViT). This demonstrates that our reflection-aware triplet mining loss is effective in detecting reflective surfaces and encourages the model to obtain accurate depth on these surfaces as shown in Figure 3. Additionally, our multi-stage approach, which employs reflection-aware knowledge distillation (denoted as *Ours*$^\dagger$), delivers comparable results across all backbone models of 3D Distillation. Note that the proposed method does not require complex scene reconstruction procedures such as mesh rendering (Matl, 2019; Newcombe et al., 2011) or ensembles of multiple neural network models (Lakshminarayanan et al., 2017).

**ScanNet-NoReflection dataset.** Table 2 summarizes the results of a quantitative evaluation performed on the ScanNet-NoReflection dataset. This evaluation aims to measure the generalization performance of models trained on datasets that include reflective surfaces. In an end-to-end training scheme, *Ours* achieves performance comparable to or within an acceptable margin of self-supervised methods. This confirms that our proposed reflection-aware triplet mining loss effectively prevents the incorrect back-propagation of the photometric loss gradient on reflective surfaces, as illustrated in Figure 2. Furthermore, the model trained by our reflection-aware knowledge distillation (*i.e.*, *Ours*$^\dagger$) shows a noticeable performance improvement, which is comparable performance to the 3D distillation method. These results suggest that extending our reflection-aware triplet mining loss to distillation techniques offers a straightforward yet effective strategy for managing reflective surfaces.

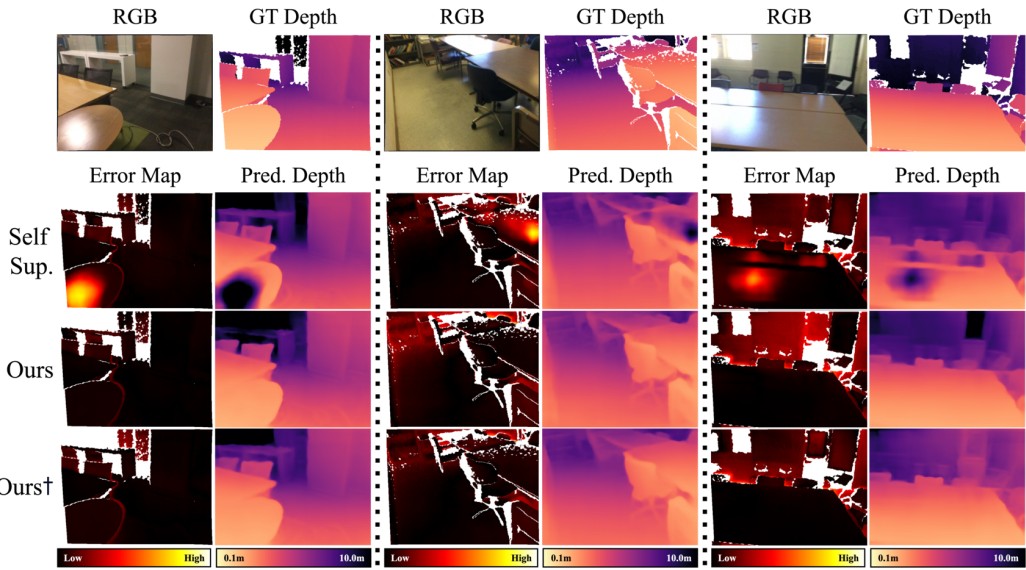

Figure 3: Qualitative results of the proposed methods on the ScanNet. We visualize the predicted depth of the Monodepth2 (Godard et al., 2019) trained by three different methods including the proposed method: Self-supervised, *Ours* and *Ours*[†]. Note that the error map represents the absolute difference between prediction and ground truth depth, normalized to between 0 and 255.

Table 3: Main results on the KITTI and NYU-v2 datasets.

| | Backbone | Method | Abs Rel ↓ | Sq Rel ↓ | RMSE ↓ | RMSE log ↓ | $\delta < 1.25$ ↑ | $\delta < 1.25^2$ ↑ | $\delta < 1.25^3$ ↑ |
|---|---|---|---|---|---|---|---|---|---|
| **KITTI** | Monodepth2 | Self-Supervised | 0.118 | 0.908 | 4.919 | 0.196 | **0.871** | 0.957 | 0.980 |
| | | *Ours* | 0.118 | 0.912 | 4.943 | 0.198 | 0.867 | 0.956 | 0.980 |
| | | *Ours*[†] | **0.116** | **0.856** | **4.796** | **0.194** | 0.870 | **0.958** | **0.981** |
| **NYUv2** | Monodepth2 | Self-Supervised | 0.171 | 0.144 | 0.622 | 0.213 | 0.746 | 0.941 | 0.985 |
| | | *Ours* | 0.166 | 0.139 | 0.616 | 0.209 | 0.759 | 0.943 | 0.985 |
| | | *Ours*[†] | **0.155** | **0.121** | **0.573** | **0.196** | **0.782** | **0.951** | **0.988** |

**KITTI and NYU-v2 datasets.**  Table 3 presents the quantitative results of the Monodepth2 architecture on the KITTI and NYU-v2 datasets, both widely used for depth estimation. In the KITTI dataset, most scenes share similar attributes and predominantly consist of non-reflective objects, limiting performance improvements in reflective regions. However, as highlighted in our study, our selective triplet mining loss using $M_r$ and the distillation process effectively preserves performance in non-reflective areas. Consequently, our method achieves performance comparable to existing approaches with negligible margins. These results are consistent with the trends observed in the ScanNet-NoReflection validation split experiments presented in Table 2. On the other hand, in the NYU-v2 dataset, where reflections are more prevalent, our triplet mining loss and distillation method yield significant performance gains. These results demonstrate that the broad applicability of our approach to indoor environments with complex light-object interactions, ensuring consistent enhancements even when incorporating a pose network.

## 4.2 CROSS-DATASET GENERALIZABILITY

To demonstrate the generalization ability across different datasets, we conduct a zero-shot evaluation using 7-Scenes and Booster datasets. As shown in Table 4, our proposed methods (denoted as *Ours* and *Ours*[†]) consistently enhances performance. Specifically, across all backbone architectures and all metrics, *Ours*[†] improved by an average of 5.47% and 13.89% for the 7-Scenes and Booster datasets, respectively. Exceptionally, there is no significant difference between *Ours* and the self-supervised method on the 7-Scenes dataset. This may be attributed to the predominance of

Table 4: Cross-dataset evaluation result on the 7-Scenes and Booster datasets.

| | Backbone | Method | Abs Rel↓ | Sq Rel↓ | RMSE↓ | RMSE log↓ | $\delta < 1.25$↑ | $\delta < 1.25^2$↑ | $\delta < 1.25^3$↑ |
|---|---|---|---|---|---|---|---|---|---|
| **7-Scenes** | Monodepth2 | Self-Supervised | 0.210 | 0.130 | 0.445 | 0.248 | 0.656 | 0.906 | 0.974 |
| | | *Ours* | 0.207 | 0.125 | 0.441 | 0.248 | 0.656 | 0.904 | 0.975 |
| | | *Ours*† | **0.198** | **0.110** | **0.415** | **0.238** | **0.667** | **0.911** | **0.980** |
| | HRDepth | Self-Supervised | 0.193 | 0.115 | 0.421 | 0.231 | 0.682 | 0.921 | 0.982 |
| | | *Ours* | 0.195 | 0.109 | 0.419 | 0.232 | 0.674 | 0.921 | 0.984 |
| | | *Ours*† | **0.183** | **0.096** | **0.389** | **0.219** | **0.706** | **0.931** | **0.986** |
| | MonoViT | Self-Supervised | 0.173 | 0.093 | 0.365 | 0.201 | 0.752 | 0.945 | 0.988 |
| | | *Ours* | 0.175 | 0.090 | 0.361 | 0.204 | 0.746 | 0.944 | 0.987 |
| | | *Ours*† | **0.162** | **0.077** | **0.335** | **0.191** | **0.776** | **0.951** | **0.989** |
| **Booster** | Monodepth2 | Self-Supervised | 0.520 | 0.429 | 0.601 | 0.444 | 0.305 | 0.591 | 0.827 |
| | | *Ours* | 0.430 | 0.301 | 0.501 | 0.389 | 0.362 | 0.675 | 0.893 |
| | | *Ours*† | **0.419** | **0.288** | **0.487** | **0.381** | **0.370** | **0.678** | **0.897** |
| | HRDepth | Self-Supervised | 0.495 | 0.391 | 0.559 | 0.426 | 0.307 | 0.611 | 0.852 |
| | | *Ours* | **0.414** | **0.276** | **0.482** | **0.379** | 0.364 | **0.680** | **0.907** |
| | | *Ours*† | 0.429 | 0.292 | 0.487 | 0.385 | **0.366** | 0.659 | 0.878 |
| | MonoViT | Self-Supervised | 0.418 | 0.327 | 0.504 | 0.374 | **0.425** | 0.679 | 0.888 |
| | | *Ours* | 0.408 | 0.302 | 0.482 | 0.362 | 0.414 | 0.677 | 0.916 |
| | | *Ours*† | **0.375** | **0.249** | **0.440** | **0.337** | 0.422 | **0.734** | **0.944** |

non-reflective surfaces in the 7-Scenes dataset, where our model, trained with the reflection-aware triplet mining loss, slightly loses high-frequency details on non-reflective surfaces. Conversely, the consistent performance improvement of *Ours*† across both reflective and non-reflective surfaces demonstrates the robust generalization capabilities of our method based on reflective region selection.

## 5 CONCLUSION

This paper addresses the intricate challenge of self-supervised monocular depth estimation on reflective surfaces. Our method employs a novel metric learning approach, centered around a reflection-aware triplet mining loss. This novel loss function significantly improves depth prediction accuracy by accurately identifying reflective regions on a per-pixel basis and effectively adjusting the minimization of photometric errors, which are typically problematic on reflective surfaces. It also preserves high-frequency details on non-reflective surfaces by selectively regulating photometric error minimization based on reflection region selection. Moreover, we introduce a reflection-aware knowledge distillation method, enabling a student model to enhance performance in both reflective and non-reflective surfaces. This method leverages the strengths of different teaching networks to produce a more robust and versatile student model. Experimental evaluations conducted on the indoor scene datasets demonstrate our method consistently enhances depth performance across various architectural frameworks. These results underscore the robustness and versatility of our approach, marking it as a valuable contribution to the field of self-supervised monocular depth estimation.

## ACKNOWLEDGEMENT

This work was supported by Korea Research Institute for defense Technology planning and advancement through Defense Innovation Vanguard Enterprise Project, funded by Defense Acquisition Program Administration (R230206) and the National Research Foundation of Korea (NRF) grant funded by the Korea government (MSIT) (No. RS-2023-00210908).

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

# A  MORE DETAILED EXPERIMENTAL SETUPS

As aforementioned in the main manuscripts, we follow all training details and experimental setups mentioned in 3D Distillation (Shi et al., 2023). We train all models with the reflection triplet split proposed by 3D Distillation for 41 epochs through the Adam optimizer (Kingma & Ba, 2014) with an image resolution of $384\times288$, implemented in PyTorch. The training batch sizes of the Monodepth2 (Godard et al., 2019), HRDepth (Lyu et al., 2021), and MonoViT (Zhao et al., 2022) are $\{12, 12, 8\}$, respectively. The initial learning rate is $10^{-4}$, and we adopt the multi-step learning rate scheduler that decays the learning rate by $\gamma = 0.1$ once the number of epochs reaches one of the milestones [26, 36]. Moreover, with 3D Distillation, the pose between cameras is ground truth during training, and the minimum and maximum depths used for training and evaluation are 0.1m and 10m. In our evaluation, we do not apply post-processing techniques such as averaging the estimates of both the flipped and original images or using median scaling.

Table 5: Main results on the ScanNet-Original Test and Validation sets.

| Backbone | Training Scheme | Method | ScanNet-Original Test Set | | | | | | |
| | | | Abs Rel ↓ | Sq Rel ↓ | RMSE ↓ | RMSE log ↓ | $\delta < 1.25$ ↑ | $\delta < 1.25^2$ ↑ | $\delta < 1.25^3$ ↑ |
|---|---|---|---|---|---|---|---|---|---|
| Monodepth2 | End-to-End | Self-Supervised | 0.189 | 0.116 | 0.407 | **0.217** | **0.731** | 0.921 | 0.974 |
| | | *Ours* | **0.185** | **0.109** | **0.405** | **0.217** | 0.730 | **0.923** | **0.975** |
| | Multi-Stage | Self-Teaching | 0.184 | 0.109 | 0.392 | 0.210 | 0.742 | 0.925 | 0.976 |
| | | 3D Distillation | 0.181 | 0.105 | 0.388 | 0.208 | **0.746** | 0.927 | 0.976 |
| | | *Ours*† | **0.175** | **0.098** | **0.385** | **0.206** | 0.746 | **0.930** | **0.979** |
| HRDepth | End-to-End | Self-Supervised | **0.184** | 0.111 | 0.399 | **0.212** | **0.739** | 0.925 | 0.976 |
| | | *Ours* | 0.186 | **0.106** | **0.397** | 0.213 | 0.735 | **0.927** | **0.977** |
| | Multi-Stage | Self-Teaching | 0.178 | 0.102 | 0.381 | 0.204 | 0.752 | 0.931 | 0.979 |
| | | 3D Distillation | 0.176 | 0.098 | 0.378 | **0.202** | 0.754 | 0.932 | 0.979 |
| | | *Ours*† | **0.173** | **0.096** | **0.375** | **0.202** | **0.755** | **0.934** | **0.980** |
| MonoViT | End-to-End | Self-Supervised | **0.154** | 0.082 | **0.343** | **0.182** | **0.801** | **0.948** | **0.984** |
| | | *Ours* | 0.155 | **0.081** | 0.345 | 0.185 | 0.795 | 0.945 | **0.984** |
| | Multi-Stage | Self-Teaching | 0.152 | 0.081 | 0.329 | 0.177 | 0.811 | 0.948 | 0.983 |
| | | 3D Distillation | **0.149** | **0.075** | **0.324** | **0.174** | **0.812** | 0.949 | **0.985** |
| | | *Ours*† | **0.149** | **0.075** | 0.335 | 0.179 | 0.805 | **0.949** | 0.980 |
| Backbone | Training Scheme | Method | ScanNet-Original Val. Set | | | | | | |
| | | | Abs Rel ↓ | Sq Rel ↓ | RMSE ↓ | RMSE log ↓ | $\delta < 1.25$ ↑ | $\delta < 1.25^2$ ↑ | $\delta < 1.25^3$ ↑ |
| Monodepth2 | End-to-End | Self-Supervised | 0.167 | 0.100 | 0.385 | 0.203 | 0.764 | 0.935 | 0.981 |
| | | *Ours* | **0.162** | **0.090** | **0.378** | **0.201** | **0.765** | **0.937** | **0.983** |
| | Multi-Stage | Self-Teaching | 0.160 | 0.090 | 0.365 | 0.193 | 0.780 | 0.941 | 0.983 |
| | | 3D Distillation | 0.157 | 0.083 | **0.357** | **0.190** | 0.782 | 0.943 | **0.985** |
| | | *Ours*† | **0.153** | **0.080** | 0.358 | **0.190** | **0.783** | **0.944** | **0.985** |
| HRDepth | End-to-End | Self-Supervised | 0.166 | 0.100 | 0.381 | 0.200 | 0.771 | 0.937 | 0.982 |
| | | *Ours* | **0.160** | **0.089** | **0.373** | **0.197** | **0.772** | **0.941** | **0.984** |
| | Multi-Stage | Self-Teaching | 0.159 | 0.090 | 0.360 | 0.190 | 0.785 | 0.943 | 0.984 |
| | | 3D Distillation | 0.154 | 0.080 | **0.349** | **0.186** | 0.788 | 0.945 | 0.986 |
| | | *Ours*† | **0.151** | **0.078** | 0.350 | **0.186** | **0.790** | **0.948** | **0.987** |
| MonoViT | End-to-End | Self-Supervised | 0.138 | 0.077 | 0.331 | **0.171** | **0.831** | 0.955 | 0.986 |
| | | *Ours* | **0.137** | **0.069** | **0.328** | 0.172 | 0.826 | **0.958** | **0.989** |
| | Multi-Stage | Self-Teaching | 0.133 | 0.071 | 0.314 | 0.163 | 0.844 | 0.959 | 0.988 |
| | | 3D Distillation | **0.128** | **0.060** | **0.296** | **0.157** | **0.846** | **0.962** | **0.990** |
| | | *Ours*† | 0.129 | 0.062 | 0.310 | 0.163 | 0.840 | 0.961 | **0.990** |

# B    EVALUATIONS ON SCANNET DATASET

To demonstrate the generalizability of our proposed method, we conduct the experiment on several ScanNet (Dai et al., 2017) splits denoted as ScanNet-Original {Test, Val.} sets and ScanNet-Robust Test set following Shi et al. (2023) and Fu et al. (2018); Bae et al. (2022), respectively. ScanNet-Original sets include both reflective and non-reflective surfaces, it is well-suited to evaluate the impact of reflective surfaces on training comprehensively. In addition, the ScanNet-Robust test set was used to measure the generalization performance of Monocular Depth Estimation in the Robust Vision Challenge 2018 (Geiger et al., 2018), as it is small-scale but suitable for evaluating generalization performance.

## B.1    EVALUATION ON SCANNET-ORIGINAL SETS

Table 5 summarizes the quantitative evaluation results of the ScanNet-Original sets. We achieve steady performance improvement across most metrics for all backbone models in the end-to-end training scheme, suggesting that the proposed method minimizes the influence of reflective surfaces, which contributes to the general depth estimation performance improvement. Furthermore, Our multi-stage training scheme (*i.e.*, *Ours*$^\dagger$) dramatically elevates performance across various depth estimation models. For Monodepth2, *Ours*$^\dagger$ achieves a remarkable average increase of 5.28% on the test set and 6.52% on the validation set across all metrics. HRDepth reaps substantial benefits, with improvements of 4.83% on the test set and 7.19% on the validation set. Likewise, MonoViT consistently gains, with enhancements of 2.28% on the test set and 5.59% on the validation set. When benchmarked against 3D Distillation (Shi et al., 2023), *Ours*$^\dagger$ provides an enhanced performance for Monodepth2, showing an average increase of 1.76% on the test set and 0.87% on the validation set. HRDepth also gains an average of 0.71% on the test set and 0.69% on the validation set. However, for MonoViT, *Ours*$^\dagger$ shows a slight decline, with decreases of 1.09% on the test set and 1.92% on the validation set compared to 3D Distillation.

Table 6: Main results on the ScanNet-Robust Test set.

| Backbone | Method | ScanNet-Robust Test Set | | | | | | |
|---|---|---|---|---|---|---|---|---|
| | | Abs Rel ↓ | Sq Rel ↓ | RMSE ↓ | RMSE log ↓ | $\delta < 1.25$ ↑ | $\delta < 1.25^2$ ↑ | $\delta < 1.25^3$ ↑ |
| Monodepth2 | Self-Supervised | 0.193 | 0.118 | 0.395 | 0.219 | 0.729 | 0.921 | 0.973 |
| | *Ours* | 0.186 | 0.107 | 0.388 | 0.216 | 0.729 | 0.926 | 0.976 |
| | *Ours*$^\dagger$ | **0.179** | **0.099** | **0.371** | **0.207** | **0.744** | **0.930** | **0.978** |
| HRDepth | Self-Supervised | 0.190 | 0.112 | 0.387 | 0.216 | 0.729 | 0.924 | 0.976 |
| | *Ours* | 0.188 | 0.107 | 0.384 | 0.215 | 0.731 | 0.926 | 0.976 |
| | *Ours*$^\dagger$ | **0.177** | **0.095** | **0.362** | **0.203** | **0.750** | **0.935** | **0.979** |
| MonoViT | Self-Supervised | 0.158 | 0.082 | 0.328 | 0.181 | 0.799 | 0.948 | 0.984 |
| | *Ours* | 0.155 | 0.078 | 0.327 | 0.183 | 0.798 | 0.949 | 0.985 |
| | *Ours*$^\dagger$ | **0.150** | **0.073** | **0.319** | **0.178** | **0.806** | **0.952** | **0.986** |

## B.2    EVALUATION ON SCANNET-ROBUST TEST SET

Table 6 summarizes the quantitative evaluation results of the ScanNet-Robust test set. Due to the 3D-distillation baselines (Shi et al., 2023) did not release the source code, and the reported performance on this split not existing, we compare the models trained by our methods to the self-supervised methods across three backbones, similar to previous experiments. As depicted in Table 6, our proposed methods (*i.e.*, *Ours*, *Ours*$^\dagger$) achieve significant performance gains for all evaluation metrics and all backbones, consistently. Specifically, for Monodepth2, *Ours* and *Ours*$^\dagger$ demonstrate an average performance improvement of 2.42% and 5.49%, respectively, across all metrics. Similarly, for HRDepth, *Ours* showed an average improvement of 1.03%, and *Ours*$^\dagger$ achieved a 5.55% increase in performance across all metrics. In the case of MonoViT, *Ours* resulted in an average performance improvement of 0.87%, and *Ours*$^\dagger$ achieved a 3.13% improvement across all metrics. The consistent improvements across all metrics for Monodepth2, HRDepth, and MonoViT indicate that our methods effectively mitigate the risk of erroneous learning induced by reflective surfaces.

Table 7: Evaluation results on the ScanNet-Reflection Test set for the combination of state-of-the-art self-supervised monocular depth framework and our approach.

| Backbone | Method | Abs Rel ↓ | Sq Rel ↓ | RMSE ↓ | RMSE log ↓ | $\delta < 1.25$ ↑ | $\delta < 1.25^2$ ↑ | $\delta < 1.25^3$ ↑ |
|---|---|---|---|---|---|---|---|---|
| RA-Depth | Self-Supervised | 0.161 | 0.124 | 0.477 | 0.203 | 0.779 | 0.947 | 0.981 |
| | RA-Depth $+\mathcal{L}_{tri}$ | 0.138 | 0.091 | 0.444 | 0.186 | 0.804 | 0.963 | 0.987 |
| | RA-Depth $+\mathcal{L}_{tri} + \mathcal{L}_{rkd}$ | **0.130** | **0.076** | **0.402** | **0.171** | **0.834** | **0.966** | **0.990** |
| GasMono | Self-Supervised | 0.156 | 0.123 | 0.462 | 0.198 | 0.810 | 0.949 | 0.980 |
| | GasMono $+\mathcal{L}_{tri}$ | 0.139 | 0.089 | 0.425 | 0.178 | 0.827 | 0.964 | 0.989 |
| | GasMono $+\mathcal{L}_{tri} + \mathcal{L}_{rkd}$ | **0.127** | **0.072** | **0.386** | **0.164** | **0.843** | **0.968** | **0.993** |
| Lite-mono | Self-Supervised | 0.179 | 0.172 | 0.517 | 0.221 | 0.775 | 0.935 | 0.973 |
| | Lite-mono $+\mathcal{L}_{tri}$ | 0.159 | 0.113 | 0.462 | 0.201 | 0.775 | 0.947 | 0.983 |
| | Lite-mono $+\mathcal{L}_{tri} + \mathcal{L}_{rkd}$ | **0.148** | **0.101** | **0.433** | **0.190** | **0.788** | **0.960** | **0.984** |

# C EXTENSIBILITY ANALYSIS OF THE PROPOSED METHOD

To demonstrate performance improvements not only across different architectures but also by applying our method while preserving state-of-the-art methodologies, we evaluate the three outperforming SSMDE models, including RA-Depth (He et al., 2022), GasMono (Zhao et al., 2023), Lite-mono (Zhang et al., 2023), which have own unique methods (*e.g.*, iterative self-distillation in GasMono). We apply our reflection-aware triplet mining loss (denoted as [Baseline] + $\mathcal{L}_{tri}$, and reflection-aware knowledge distillation method (denoted as [Baseline] + $\mathcal{L}_{tri} + \mathcal{L}_{rkd}$) to each baseline. Table 7 summarizes the results of the ScanNet-Reflection test split, comparing the performance of each method when incorporating the triplet loss and our distillation method. The results demonstrate that our method provides substantial performance gains even for recent SoTA methods, underscoring its robustness to reflective surfaces.

Table 8: Evaluation results on the ScanNet-Reflection Validation and ScanNet-NoReflection Validation sets. w.r.t. reflective region mask $M_r$.

| Backbone | Method | ScanNet-Reflection Validation Set | | | | ScanNet-NoReflection Validation Set | | | |
|---|---|---|---|---|---|---|---|---|---|
| | | Abs Rel ↓ | Sq Rel ↓ | RMSE ↓ | $\delta < 1.25^3$ ↑ | Abs Rel ↓ | Sq Rel ↓ | RMSE ↓ | $\delta < 1.25^3$ ↑ |
| Monodepth2 | $M_r = 0$ | 0.206 | 0.227 | 0.584 | 0.961 | 0.169 | 0.100 | **0.395** | 0.979 |
| | $M_r = 1$ | 0.170 | 0.132 | 0.505 | 0.979 | 0.171 | 0.099 | 0.402 | 0.978 |
| | *Ours* | **0.166** | **0.125** | **0.492** | **0.981** | **0.168** | **0.095** | **0.395** | **0.980** |
| HRDepth | $M_r = 0$ | 0.213 | 0.244 | 0.605 | 0.961 | 0.169 | 0.102 | **0.388** | **0.980** |
| | $M_r = 1$ | 0.184 | 0.167 | 0.564 | 0.965 | 0.179 | 0.113 | 0.433 | 0.968 |
| | *Ours* | **0.167** | **0.127** | **0.496** | **0.982** | **0.167** | **0.096** | 0.389 | 0.979 |
| MonoViT | $M_r = 0$ | 0.179 | 0.206 | 0.557 | 0.963 | **0.140** | 0.074 | **0.333** | 0.984 |
| | $M_r = 1$ | 0.155 | 0.151 | 0.527 | 0.971 | 0.168 | 0.112 | 0.420 | 0.954 |
| | *Ours* | **0.139** | **0.107** | **0.452** | **0.984** | 0.141 | **0.072** | 0.338 | **0.987** |

# D IMPACT OF THE REFLECTION-AWARE TRIPLET MINING LOSS W.R.T. REFLECTIVE REGION LOCALIZATION

As aforementioned in the main manuscript, the proposed reflection-aware triplet mining loss is applied to reflective regions, thus preserving performance in non-reflective regions. To validate this claim, we conduct an experiment to evaluate the impact of varying the reflection mask $M_r$ with three configurations as follows:

1. $M_r = 0$: This configuration exactly corresponds to the traditional self-supervised method without the triplet mining loss.

2. $M_r = 1$: In this configuration, the triplet loss is applied to both reflective and non-reflective regions of the image.

3. *Ours*: This configuration leverages $M_r$, which is calculated through Equation 8 in the main manuscript, to selectively regulate the reflective regions.

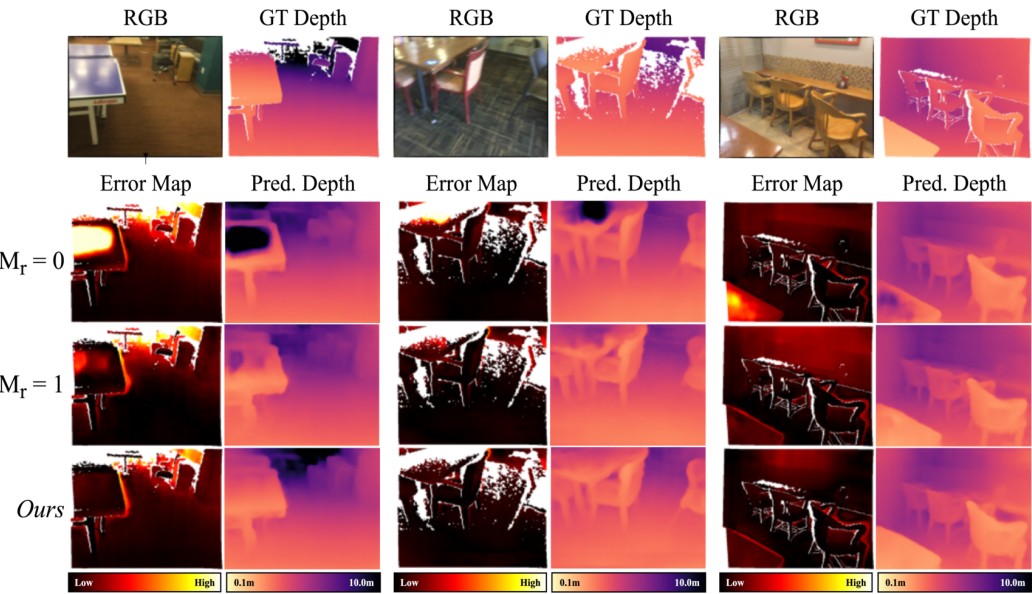

Figure 4: Qualitative results of the proposed methods w.r.t. reflective region mask $M_r$.

As shown in Table 8, the results demonstrate that *Ours* significantly improves performance on reflective datasets while maintaining comparable performance on non-reflective regions when compared to the first configuration (denoted as $M_r = 0$). On the other hand, applying the triplet mining loss across all regions ($M_r = 1$) led to some performance improvement in reflective regions but resulted in a notable drop in performance in non-reflective regions compared to other configurations. These findings verify that the proposed reflection-aware triplet mining loss effectively identifies reflective regions and applies the triplet loss selectively, thereby preserving the performance in non-reflective regions.

Table 9: Computational overhead comparison.

| Method | | Task | Training cost (hours) |
|---|---|---|---|
| Self-supervised | **(1)** | End-to-end training | 11.5 |
| Ours (triplet loss) | **(2)** | End-to-end training | 14.1 |
| Ours (distillation) | **(3)** | Multi-stage training | 27.3 |
| 3D Distillation | **(4)** | Total (1+2+3+4) | 95.5 |
| | | 1. Ensemble model pre-training | 65.8 |
| | | 2. Ensemble model inference time | 4.8 |
| | | 3. Mesh reconstruction and rendering | 11.7 |
| | | 4. Student model training | 13.2 |

## E    COMPUTATIONAL OVERHEAD COMPARISON WITH 3D DISTILLATION

The proposed triplet mining loss has a negligible impact on the training cost of the model. Although the proposed simple distillation method involves multi-stage training and incurs additional training costs, it remains significantly more efficient than the existing 3D distillation approach (Shi et al., 2023). To provide clarity, we summarized the training costs of Monodepth2 (Godard et al., 2019) under four scenarios: **(1)** training with traditional self-supervision, **(2)** applying our triplet loss, **(3)** applying our proposed distillation method, and **(4)** employing the 3D distillation approach. All training times were measured using a single *RTX A6000* GPU, as detailed in the Table 9.

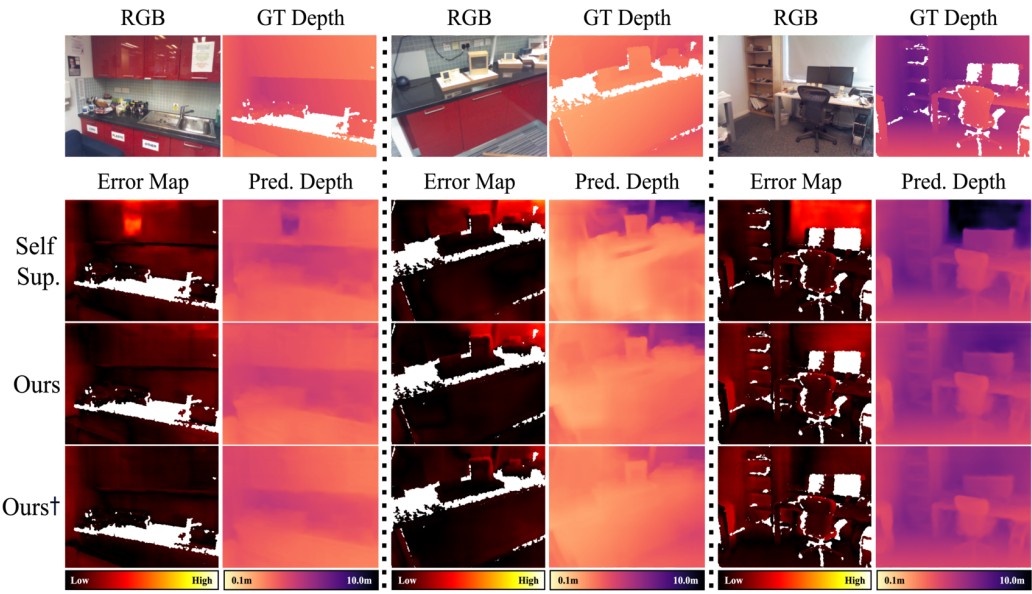

(a) Qualitative results of the proposed methods on the 7-Scenes dataset.

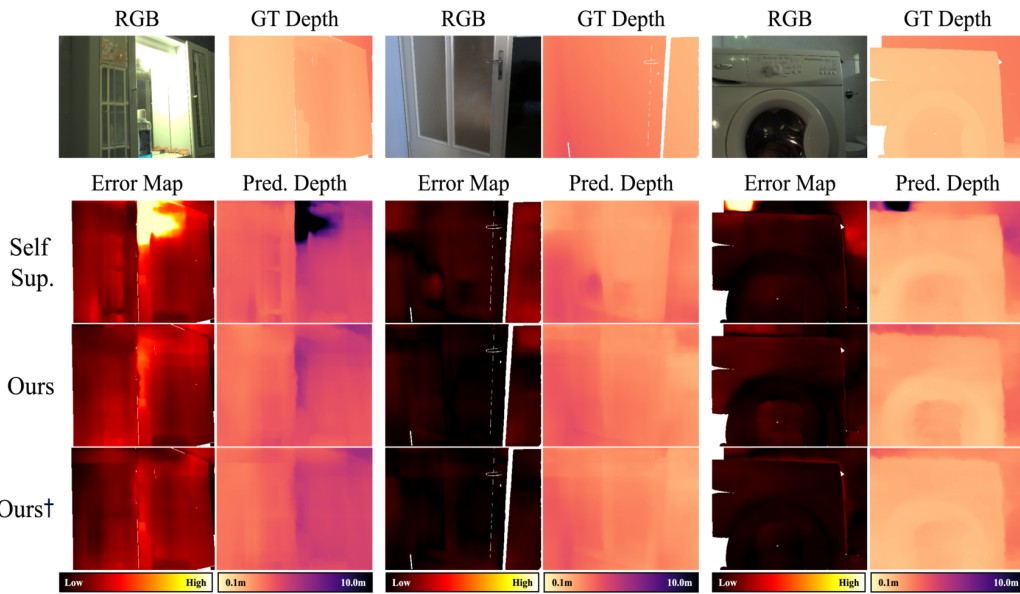

(b) Qualitative results of the proposed methods on the Booster dataset.

Figure 5: Qualitative results of the proposed methods on the 7-scenes and Booster datasets. Note that the error map represents the absolute difference between prediction and ground truth depth, normalized to between 0 and 255.

## F  QUALITATIVE RESULTS ON 7-SCENES AND BOOSTER DATASETS

We provide additional qualitative results of the proposed methods denoted as *Ours* and *Ours*[†] as discussed in Table 4 of the main manuscript, utilizing the 7-Scenes (Shotton et al., 2013) and Booster (Ramirez et al., 2023) datasets. In Figure 5, we showcase the predicted depth and error map of the Monodepth2 trained by self-supervised, $\mathcal{L}_{tri}$, and $\mathcal{L}_{rkd}$. Our methods alleviate the black-hole effect on specular highlight regions while preserving high-frequency details on non-reflective areas.

As demonstrated by the qualitative evaluation of the 7-Scenes dataset, our proposed methods exhibit robustness to reflective surfaces and impressive performance in preserving details on non-reflective surfaces in other indoor scenes that are similar to the training environment.

## G    LIMITATIONS

Despite the promising results, our study has several limitations. One major limitation is that the proposed method cannot handle transparent or mirror (ToM) objects. Secondly, a few cases do not satisfy the assumption of Equation 8 of the manuscript (*e.g.*, surfaces including multiple reflection lobes). Lastly, similar to 3D distillation (Shi et al., 2023), the conducted experiments assume that the ground truth camera pose is known during training. Future research should aim to address this limitation by exploring more effective solutions. We encourage future researchers to further investigate this issue and develop improved methodologies to overcome these challenges.

