# OpenReview forum: "Self-supervised Monocular Depth Estimation Robust to Reflective Surface Leveraged by Triplet Mining"
_ICLR.cc/2025/Conference — ICLR 2025 Poster_

### Official Review · Reviewer_tqa3 · 2024-10-29

**Soundness:** 2
**Presentation:** 3
**Contribution:** 2
**Rating:** 6
**Confidence:** 5

**Summary:**

This paper aims to address inaccurate depth estimation on the reflective surfaces in a scene. To this end, this paper proposes a training strategy for self-supervised depth estimation by leveraging triplet mining. The key idea is to penalize the inappropriate photometric error minimization on the localized reflective regions while preserving depth accuracy on non-reflective areas. In addition, this paper introduces a reflection-aware knowledge distillation method that enables a student model to selectively learn the pixel-level knowledge from reflective and non-reflective regions. Finally, the proposed framework is validated on various indoor datasets.

**Strengths:**

1.	This paper is well-written and the proposed method is easy to comprehend.
2.	This method is simple and general, which can be used for most self-supervised depth estimation methods.

**Weaknesses:**

1.Whether Eq. (8) can accurately reflect the reflective surface? Due to the ambiguity of photometric errors, the magnitude of photometric errors is not even a measure of the accuracy of depth [1], let alone reflective surfaces .
2.The overall performance is very marginal. From the results on the test set in Table 4, we can find that, like most end-to-end self-supervised training solutions, all comparison methods have limited improvement or even worse after using the idea proposed by the author.
3.Experimental verification is inadequate. The selected baselines, Monodepth2(2019), HRDepth(2021), MonoViT(2020) are the early works, not the SOTA framework. I am concerned about the effectiveness of the triplet mining loss when incorporated into recent works. Moreover, there is a lack of comparison on common depth estimation datasets, such as KITTI, NYUv2, etc., where reflective surfaces also exist.

[1] Frequency-aware self-supervised monocular depth estimation.

**Questions:**

1.	Verify effectiveness on some recent SOTA frameworks, such as [1,2,3]
2.	Training and validation on common self-supervised depth estimation datasets like KITTI, NYUv2.
3.	See weaknesses for other issues.

[1] RA-Depth: Resolution Adaptive Self-Supervised Monocular Depth Estimation.
[2] GasMono: Geometry-Aided Self-Supervised Monocular Depth Estimation for Indoor Scenes.
[3] Self-supervised Monocular Depth Estimation with Large Kernel Attention.

---

> ### Author Response · Authors · 2024-11-20
> **Response to Reviewer tqa3**
>
> >_**W1.**_ Can the reflection region mask detect reflective surfaces solely based on photometric error?
>
> >_**A1.**_ As you pointed out, detecting reflective surfaces accurately based solely on photometric error is indeed a challenging task. Therefore, we aim to explain this based on three main points, both theoretically and experimentally, that we have observed:
>
> > **Subject 1. How can Equation 8 theoretically localize reflective objects (or surfaces)?**
> >
> >As mentioned in the main manuscript, our method identifies reflective regions based on the characteristic behavior of disparities on reflective surfaces. These disparities are abnormally smaller than those on actual surfaces when observed across different camera coordinate systems (_i.e.,_ reference camera coordinate ↔ source camera coordinate). This observation is well-established (please refer to response **A2** of **Reviewer ptjG** for further clarification). To summarize our method procedure qualitatively, it is as follows:
>
> > * Assumption: When the depth network is trained with sufficient accuracy, the photometric error between $I_{s2r}$ and $I_{ref}$​, aligned in the reference camera coordinate (denoted as $E^+$ in ```Equation 8```), should be smaller than the photometric error between $I_{s2r}$ and $I_{r2s}$​, in different camera coordinates (denoted as $E^-$ in ```Equation 8```).
> > * On reflective surfaces, the abnormally low disparity of mirrored objects leads to a significantly smaller photometric error between different camera coordinates ($E^-$).
> > * Therefore, when the photometric error between different camera coordinates  $E^-$ is smaller than the photometric error between images aligned in the same camera coordinate $E^+$ by a certain margin $\delta$ (indicating a violation of the upper assumption), these regions are considered a reflective region.
>
> >As you mentioned, due to photometric ambiguity, not only reflective objects but also some non-reflective surfaces can violate the above assumption and be localized by the reflective region mask $M_r$​. For example, we observe that textureless surfaces or surfaces with multiple reflective lobes can be localized by ```Equation 8```. However, as we will describe in **Subject 2**, the triplet mining loss during the training process can adequately address these issues.
>
> > **Subject 2. How does our triplet mining loss operate during the training process for localized reflective regions?**
> >
> >In summary, our triplet mining loss functions as a regularization term for photometric error minimization.
> As depicted in Figure 2 of the main manuscript, our triplet loss has a dual effect: it acts as a regularization mechanism by maximizing $E^-$, actively increasing the abnormally low disparities in different camera coordinates while simultaneously counterbalancing the erroneous photometric error minimization of $E^+$.
>
> >Additionally, though not explicitly mentioned in the main manuscript, we adopt the smoothness term proposed in Monodepth2 [1] to ensure training stability. The core concept of the smoothness term is to encourage the propagation of similar depth values to pixels with similar colors, thereby promoting spatial consistency. This helps ensure that depth predictions remain coherent across neighboring regions, especially in areas prone to photometric ambiguities. The smoothness term plays a critical role in stabilizing the training process, as its relative contribution increases in such cases, effectively rectifying these errors and enhancing overall robustness.
>
> >Given that most objects exhibit sufficiently wide planar surfaces with continuous characteristics, except at edges, this approach also mitigates the impact of mistakenly localized regions, such as textureless surfaces or reflection lobes, as described in **Subject 1**. Consequently, these mislocalized regions have minimal influence on the overall training process, ensuring the robustness and reliability of our method.
>
> >**Subject 3. How did we experimentally validate the effectiveness of the reflective region mask?**
> >
> >First, we demonstrated the effectiveness of our reflective region mask $M_r$, indirectly through ablation studies presented in the supplementary material (please refer to response **A1** of **Reviewer jMZ7** for a more detailed explanation).
> Additionally, as shown in ```Figure 5``` of the supplementary material, utilizing our reflective region mask allows for accurate depth estimation in both reflective and non-reflective regions.
> This result indicates that the reflective regions selectively identified by $M_r$ are effectively regularized, as supported by the smoothness term described in **Subject 2**.

---

> ### Author Response · Authors · 2024-11-20
> **Continued response to Reviewer tqa3**
>
> >_**W2.**_ Limited improvement of ```Table 4``` in main manuscript (_Scannet-Original split_)
>
> >_**A2.**_ ```Table 4``` summarizes the evaluation results on the Scannet-Original dataset, which contains a low proportion of reflective surfaces and is predominantly composed of non-reflective surfaces. Our study focuses on addressing the incorrect photometric error minimization observed on reflective surfaces rather than enhancing performance on non-reflective surfaces. Consequently, the performance gains on this dataset are naturally less pronounced compared to datasets predominantly containing reflective surfaces.
>
> >To demonstrate that our method can effectively correct severe errors on reflective surfaces while preserving performance on non-reflective surfaces, we included ```Table 4``` alongside results from other datasets dominated by non-reflective surfaces (e.g., ```Table 2```, ```Table 4```, and ```Table 6``` in the main manuscript, supplementary materials, and experimental results in **A3-2** on KITTI and NYU-v2).
> Thus, rather than viewing this as _"limited improvement"_ on non-reflective datasets, our method can be interpreted as achieving significant improvements on reflective surfaces (_e.g., resolving the black-hole effect_) while maintaining comparable performance on non-reflective surfaces. This balance highlights the robustness of our approach and its ability to handle diverse surface types effectively.
>
> ---
>
> >_**W3-1, Q1**_. Old and inadequate baselines.
>
> >To ensure a fair comparison with the prior study, 3D Distillation **[4]**, we controlled the variables in the training scenario. For the main experiments, we used influential model architectures in SSMDE, such as Monodepth2 **[1]**, HRDepth **[2]**, and MonoViT **[3]**, as the variable factors. Note that MonoViT (3DV 2022) has demonstrated superior performance in quantitative evaluations, surpassing recent advancements like LiteMono (CVPR 2023) and RA-Depth (ECCV 2022), as highlighted in the robustness assessment of RoboDepth **[5]** (NeurIPS 2024). Given its consistent adoption in SSMDE and its status as a high-performing recent work, we deemed it an adequate choice for our experiments. However, we strongly agree with your suggestion that our paper would benefit significantly from demonstrating performance improvements not only across different architectures but also by applying our method while preserving state-of-the-art methodologies.
>
> >Due to the implementation code of “SSMDE with Large Kernel Attention” you mentioned is not publicly released, we incorporated experiments on three recent methods: **RA-Depth [6]** and **GasMono [7]**, and a new architectural design, **LiteMono [8]**, based on your suggestion. To simulate the training scenario proposed in the main text—_where models are continuously exposed to reflective objects during training_—we used the ScanNet-Reflection training split for these experiments.
> Since the architectural designs of RA-Depth and GasMono are identical or similar to HRDepth and MonoViT, respectively, we retained the core concepts proposed in their papers (_e.g.,_ iterative self-distillation in GasMono) and retrained the models accordingly. For LiteMono, we followed the same training scenario as 3D Distillation while replacing the model architecture with LiteMono.
> The table below summarizes the results of the ScanNet-Reflection test split, comparing the performance of each method when incorporating the triplet loss and our distillation method. The results demonstrate that our method provides substantial performance gains even for recent SoTA methods, underscoring its robustness to reflective surfaces.
>
> >| Method                | Abs Rel | Sq Rel | RMSE  | RMSE log | d1   | d2   | d3   |
> >|-----------------------|---------|--------|-------|----------|-------|-------|-------|
> >| RA-Depth             | 0.161   | 0.124  | 0.477 | 0.203    | 0.779 | 0.947 | 0.981 |
> >| RA-Depth + $\cal{L}_{tri}$ | 0.138   | 0.091  | 0.444 | 0.186    | 0.804 | 0.963 | 0.987 |
> >| RA-Depth + $\cal{L}_{tri}$ + distillation| **0.130**   | **0.076**  | **0.402** | **0.171**  | **0.834** | **0.966** | **0.990** |
> >| GasMono              | 0.156   | 0.123  | 0.462 | 0.198    | 0.810 | 0.949 | 0.980 |
> >| GasMono + $\cal{L}_{tri}$  | 0.139   | 0.089  | 0.425 | 0.178    | 0.827 | 0.964 | 0.989 |
> >| GasMono + $\cal{L}_{tri}$ + distillation| **0.127**   | **0.072**  | **0.386** | **0.164**    | **0.843** | **0.968** | **0.993** |
> | Lite-mono            | 0.179   | 0.172  | 0.517 | 0.221    | 0.775 | 0.935 | 0.973 |
> | Lite-mono + $\cal{L}_{tri}$| 0.159   | 0.113  | 0.462 | 0.201    | 0.775 | 0.947 | 0.983 |
> | Lite-mono + $\cal{L}_{tri}$ + distillation| **0.148**  | **0.101**  | **0.433** | **0.190**  | **0.788** | **0.960** | **0.984** |

---

> ### Author Response · Authors · 2024-11-20
> **Continued response to Reviewer tqa3**
>
> >_**W3-2, Q2**_. Lack of experiments on common depth datasets, including KITTI and NYUv2
>
> >_**A3-2.**_ Our research primarily focuses on mitigating the severe misestimation of depth in reflective regions (_i.e.,_ specular highlights, glossy effects) in self-supervised monocular depth estimation, a phenomenon often referred to as the _"black-hole effect."_ This issue is typically more pronounced in indoor scenes, where complex lighting interactions with objects are common, leading to frequent occurrences of reflections. Therefore, as suggested in previous work **[4]**, we concentrated our experiments on indoor datasets with reflection-dominant scenes rather than non-reflection-dominant scenes like the KITTI dataset. However, we agree that experiments on outdoor scenes, such as KITTI, are important since reflective objects, though uncommon, can still appear in outdoor environments (_e.g.,_ wet road surfaces and reflective building surfaces).
>
> >Following your suggestion, along with that of **Reviewer ptjG**, we tested our method on widely used datasets, including KITTI (outdoor) and NYU-v2 (indoor). Additionally, to demonstrate the effectiveness of our method in addressing reflective surface challenges in outdoor scenes, we evaluated it on the PittsStereo-RGBNIR dataset **[9]**. This outdoor dataset includes challenging reflective materials such as Light, Glass (Reflection), and Glossy (Reflection) types (To see demo images and videos: https://www.cs.cmu.edu/~ILIM/projects/AA/RGBNIRStereo/).
>
> >For the KITTI and NYU-v2 experimental setups, we followed the training protocol of **[1]**, incorporating our reflection-aware triplet loss and distillation training procedure. For the PittsStereo-RGBNIR dataset, we used RGB images exclusively for testing, as self-supervised monocular depth training was not feasible due to the absence of camera intrinsics and RGB stereo pairs. We performed a zero-shot evaluation on the PittsStereo-RGBNIR dataset using models trained on the KITTI dataset (_i.e.,_ Monodepth2 and Monodepth2 with our methods).
>
> >|Dataset|Method|Abs Rel|Sq Rel|RMSE|RMSE log|d1|d2|d3|
> >|---|---|---|---|---|---|---|---|---|
> >|KITTI|monodepth2|0.118|0.908|4.919|0.196|**0.871**|0.957|0.980|
> >|KITTI|monodepth2 + $\cal{L}_{tri}$|0.118|0.912|4.943|0.198|0.867|0.956|0.980|
> >|KITTI|monodepth2 + $\cal{L}_{tri}$ + distillation|**0.116**|**0.856**|**4.796**|**0.194**| 0.870 |**0.958**|**0.981**|
> >|NYU-v2|monodepth2|0.171|0.144|0.622|0.213|0.746|0.941|0.985|
> >|NYU-v2|monodepth2 + $\cal{L}_{tri}$ | 0.166   | 0.139  | 0.616 | 0.209| 0.759 | 0.943 | 0.985|
> >|NYU-v2|monodepth2 + $\cal{L}_{tri}$ + distillation|**0.155**|**0.121**|**0.573**|**0.196**|**0.782**|**0.951**|**0.988**|
>
> >The upper table summarizes the evaluation results of each dataset under three methods. Due to constraints on computing resources and time, we limited our evaluation to the **[1]**'s architecture across new datasets.
>
> >In the case of the KITTI dataset, most scenes exhibit similar attributes and are predominantly composed of non-reflective objects, resulting in limited performance gains in reflective regions. However, as highlighted in our paper, our selective triplet mining loss using $M_r$​ and the distillation process effectively preserved performance in non-reflective areas. Consequently, our method achieved comparable performance to existing approaches with negligible margins. These results are consistent with the trends observed in the ScanNet-NoReflection validation split experiments presented in ```Table 2``` of the main manuscript.
>
> >On the other hand, in the NYU-v2 dataset, where reflections occur more frequently, we observed meaningful performance improvements through our triplet mining loss and distillation method. These results demonstrate that our method is broadly applicable to indoor scenes characterized by complex interactions between light and diverse objects, enabling consistent performance enhancements in such scenarios.
>
> >|Dataset|Method|Avg. (RMSE) | Common|Light| Glass* | Glossy* | Vegetation|Skin| Clothing | Bag |
> >|---|---|---|---|---|---|---|---|---|---|---|
> >|PittsStereo|monodepth2 | 2.429 | 2.749| 3.505|2.763| 3.910|1.549|1.821|1.817|1.318|
> >|PittsStereo|monodepth2 + $\cal{L}_{tri}$|2.359|2.972|3.654|2.573| **3.382**| 1.390| 1.627| 1.745|1.526|
> >|PittsStereo|monodepth2 + $\cal{L}_{tri}$ + distillation | **2.070**|**2.530**| **3.077**| **2.270** | 3.493|**1.222**| **1.552**|**1.514**|**0.906**|
>
> >Evaluation of three models trained on the KITTI dataset against the PittsStereo-RGBNIR dataset revealed that our methods achieved significant performance improvement for reflective surfaces, such as **Glass**, and **Glossy** classes. Furthermore, when using our distillation method, we observed substantial performance improvements not only for reflective surfaces but also for non-reflective classes. This validates our claim that the proposed distillation method can substantially enhance the performance on non-reflective surfaces.

---

> ### Author Response · Authors · 2024-11-20
> **Continued response to Reviewer tqa3**
>
> ### **References**
>
> _[1] Godard, Clément, et al. "Digging into self-supervised monocular depth estimation." Proceedings of the IEEE/CVF international conference on computer vision. 2019._
>
> _[2] Lyu, Xiaoyang, et al. "Hr-depth: High resolution self-supervised monocular depth estimation." Proceedings of the AAAI conference on artificial intelligence. Vol. 35. No. 3. 2021._
>
> _[3] Zhao, Chaoqiang, et al. "Monovit: Self-supervised monocular depth estimation with a vision transformer." 2022 international conference on 3D vision (3DV). IEEE, 2022._
>
> _[4] Shi, Xuepeng, et al. "3D Distillation: Improving Self-Supervised Monocular Depth Estimation on Reflective Surfaces." Proceedings of the IEEE/CVF International Conference on Computer Vision. 2023._
>
> _[5] Kong, Lingdong, et al. "Robodepth: Robust out-of-distribution depth estimation under corruptions." Advances in Neural Information Processing Systems 36 (2024)._
>
> _[6] He, Mu, et al. "Ra-depth: Resolution adaptive self-supervised monocular depth estimation." European Conference on Computer Vision. Cham: Springer Nature Switzerland, 2022._
>
> _[7] Zhao, Chaoqiang, et al. "GasMono: Geometry-aided self-supervised monocular depth estimation for indoor scenes." Proceedings of the IEEE/CVF International Conference on Computer Vision. 2023._
>
> _[8] Zhang, Ning, et al. "Lite-mono: A lightweight cnn and transformer architecture for self-supervised monocular depth estimation." Proceedings of the IEEE/CVF Conference on Computer Vision and Pattern Recognition. 2023._
>
> _[9] Zhi, Tiancheng, et al. "Deep material-aware cross-spectral stereo matching." Proceedings of the IEEE conference on computer vision and pattern recognition. 2018._
>
> _PittsStereo-RGBNIR Dataset Demopage:_ https://www.cs.cmu.edu/~ILIM/projects/AA/RGBNIRStereo/

---

> ### Comment · Reviewer_tqa3 · 2024-11-26
>
> Thank you for the detailed response. This addresses most of my questions. I also appreciate the simplicity and straightforwardness of this solution as applied to self-supervised monocular depth estimation (SSDE), and I am increasing my rating to 6.

---

> > ### Author Response · Authors · 2024-11-26
> > **Gratitude for Your Feedback and Evaluation**
> >
> > We are very pleased to have been able to adequately address your concerns.
> >
> > We believe that the discussions we had with you will greatly contribute to the overall quality of the paper.
> >
> > Thank you very much for your careful evaluation and for raising the score.

---

### Official Review · Reviewer_jMZ7 · 2024-11-02

**Soundness:** 3
**Presentation:** 3
**Contribution:** 3
**Rating:** 8
**Confidence:** 5

**Summary:**

This paper proposes a novel self-supervised monocular depth estimation training strategy, which uses triplet mining loss and reflection-aware knowledge distillation to pinpoint reflective regions at the pixel level, resulting in a robust depth estimation.

**Strengths:**

1.The proposed triplet mining loss can be integrated into a general monocular depth estimation framework, enhancing the accuracy of reflective surfaces.

2.The proposed reflection-aware knowledge distillation enhances the accuracy of reflective surfaces while preserving high-frequency details of non-reflective surfaces.

3.Experiments on multiple datasets demonstrate that the proposed method achieves state-of-the-art performance.

**Weaknesses:**

1.Ablation study on the proposed pixel-level reflection regions Mr are necessary to further analyze its effectiveness in reflection-aware triplet mining loss and reflection-aware knowledge distillation.

2.Will the proposed triplet mining strategy and reflection-aware knowledge distillation introduce additional computational overhead?

3.The description in lines 246-247 is confusing. The relative pose [R|t]s2r is obtained by computing the inverse of [R|t]s2r?

4. Depth completion is an extended task from depth estimation. Thus, it would be better if adding some depth completion methods [1-5] into the related work section.

[1] Bilateral Propagation Network for Depth Completion, CVPR 2024

[2] Tri-Perspective View Decomposition for Geometry-Aware Depth Completion, CVPR 2024

[3] RigNet: Repetitive Image Guided Network for Depth Completion, ECCV 2022

[4] Self-supervised Sparse-to-Dense: Self- supervised Depth Completion from LiDAR and Monocular Camera, ICRA 2019

[5] Desnet: Decomposed Scale-Consistent Network for Unsupervised Depth Completion, AAAI 2023

**Questions:**

I would appreciate it If the authors can address my concerns in Weaknesses.

---

> ### Author Response · Authors · 2024-11-20
> **Response to Reviewer jMZ7**
>
> >_**W1.**_ Ablation study of reflection region mask.
>
> >_**A1.**_ Due to space constraints in the main manuscript, we included the ablation studies of the reflection region mask $M_r$​ in the supplementary material. We sincerely apologize for any inconvenience this may have caused.
> >The ablation studies indirectly demonstrate the effectiveness of our reflective region mask $M_r$​, in localizing reflective regions. As shown in ```Section D``` and ```Table 6``` of the supplementary material, the depth network without triplet loss resulted in a good performance in non-reflective regions but a significant performance drop in reflective areas. Conversely, applying the triplet loss indiscriminately across all regions without the mask ($M_r$)​ improved performance in reflective regions but caused substantial degradation in non-reflective regions.
>
> >In contrast, using our proposed $M_r$​, to selectively apply the triplet mining loss to reflective regions enabled the model to achieve strong performance in both reflective and non-reflective areas. This demonstrates that the combination of our reflective region mask and triplet mining loss can effectively preserve or enhance depth quality across all regions.
>
> >Additionally, we have detailed how the reflection region mask $M_r$ can detect reflective surfaces solely based on photometric error in response **A1** to **Reviewer tqa3**. If you are interested in understanding the qualitative mechanism of the reflective region mask during the training process, we encourage you to refer to that response.
>
> ---
>
> >_**W2.**_ Computational overhead comparison.
>
> >_**A2.**_ The proposed triplet mining loss has a negligible impact on the training cost of the model. Although the proposed simple distillation method involves multi-stage training and incurs additional training costs, it remains significantly more efficient than the existing 3D distillation approach **[1]**. To provide clarity, we summarized the training costs of Monodepth2 **[2]** under four scenarios: _(1) training with traditional self-supervision, (2) applying our triplet loss, (3) applying our proposed distillation method, and (4) employing the 3D distillation approach_. All training times were measured using a single RTX A6000 GPU, as detailed in the table below:
>
> >|Method|Task|Training Cost (hours)|
> >|---|---|---|
> >|**Self-sup.** (1)|End-to-end training|__11.5__|
> >|**Ours (triplet loss)** (2)|End-to-end training|__14.1__|
> >|**Ours (distillation)** (3)|Multi-stage training|__27.3__|
> >|**3D Distillation** (4)|Total (1+2+3+4)|__95.5__|
> >||1. Ensemble Model Pre-training|65.8|
> >||2. Ensemble Model Inference Time|4.8|
> >||3. Mesh Reconstruction and Rendering|11.7|
> >||4. Student Model Training|13.2|
>
> >By incorporating our triplet loss into the self-supervised method, we achieved significant performance improvements with only a negligible additional training cost compared to (1) the method training with the traditional self-supervision. Moreover, our proposed distillation method reduced the training cost of 3D distillation by 71.41% while achieving comparable performance with 3D Distillation.
>
> ---
>
> >_**W3.**_ Miswriting in ```lines 246-247```
>
> >_**A3.**_ We sincerely appreciate your careful feedback. the sentence you mentioned should be changed as follows:
>
> > * _Before:_ “where the relative pose [R|t]_{s2r} can be obtained by computing the inverse of the predicted pose ```$[R|t]_{s2r}$```”
> >
> > * _After:_ “where the relative pose [R|t]_{s2r} can be obtained by computing the inverse of the predicted pose ```$[R|t]_{r2s}$```”
>
> >We apologize for any confusion this typo may have caused during the review process. We will ensure this correction is made in the revised version of the main manuscript.
>
> ---
>
> >_**W4.**_ Lack of literature review of Depth completion.
>
> >_**A4.**_ Thank you for your constructive feedback. We will take your feedback into account and update the final version accordingly.
>
> We sincerely hope that our responses have adequately addressed your concerns and clarified the points raised. We are grateful for the time and insight you've dedicated to reviewing our work, and we look forward to any further feedback that may help us enhance the clarity and rigor of our research.
>
> ### **References**
>
> _[1] Shi, Xuepeng, et al. "3D Distillation: Improving Self-Supervised Monocular Depth Estimation on Reflective Surfaces." Proceedings of the IEEE/CVF International Conference on Computer Vision. 2023._
>
> _[2] Godard, Clément, et al. "Digging into self-supervised monocular depth estimation." Proceedings of the IEEE/CVF international conference on computer vision. 2019._

---

> ### Comment · Reviewer_jMZ7 · 2024-11-22
> **Thanks for the reply.**
>
> The reply has addressed most of my concerns. I vote for acceptance.

---

> > ### Author Response · Authors · 2024-11-22
> > **Gratitude for Your Feedback and Evaluation**
> >
> > We are very pleased to have been able to adequately address your concerns.
> >
> > We believe that the discussions we had with you will greatly contribute to the overall quality of the paper.
> >
> > Thank you very much for your careful evaluation and for raising the score.

---

### Official Review · Reviewer_ptjG · 2024-11-03

**Soundness:** 2
**Presentation:** 3
**Contribution:** 3
**Rating:** 6
**Confidence:** 4

**Summary:**

The paper proposes a reflection-aware training method for self-supervised monocular depth estimation. The proposed method leverages the triplet mining loss to regularize the inappropriate regions for photometric loss.
Key contributions of this work includes:
- Introducing reflection-aware triplet mining loss and knowledge distillation for reflective regions.
- first end-to-end method for self-supervised monocular depth estimation in reflective regions.
- diverse experimetns on various indoor datasets.

**Strengths:**

S1. This is the first end-to-end method for self-supervised monocular depth estimation in reflective regions. The proposed triplet loss is very simple and can be integrated into any framework for SSMDE.

S2. Diverse Experiments: The paper shows diverse experiments with multiple datasets and cross validations.

S3.  Writing and Presentation: The paper is well-written and easy to understand.

**Weaknesses:**

W1. Many self-supervised monocular depth estimation methods are evaluated in outdoor scenes like KITTI. However, this paper shows their results in only indoor scenes using Scannet, 7-Scenes and Booster datasets.

W2. I could not find the proof of the authors' assumption that reflective regions have low disparities (instead of larger disparities).

**Questions:**

I would like to start by thanking the authors for their contribution to this field with their submission. Here are some questions:

Q1: This relates to W1. Are there any reasons for not including experiments on outdoor scenes, such as the KITTI dataset? The proposed method is demonstrated on indoor datasets, where pose prediction networks might be challenging to train due to prevalent rotational camera movements. I am curious whether this method is adaptable when a pose network is involved or if it can be applied to depth estimation tasks in outdoor scenes.

Q2: This relates to W2. I am concerned about whether it is possible to say that the predicted depth maps for reflective regions always have small disparities (instead of randomness errors). Is there any provements or previous works showing this is true?

---

> ### Author Response · Authors · 2024-11-20
> **Response to Reviewer ptjG**
>
> >_​**​W1, Q1.**_ Evaluation of our method to KITTI dataset & Involving pose network to our method.
>
> >_**A1.**_ Our research primarily focuses on mitigating the severe misestimation of depth in reflective regions (_i.e.,_ specular highlights, glossy effects) in self-supervised monocular depth estimation, a phenomenon often referred to as the _"black-hole effect."_ This issue is typically more pronounced in indoor scenes, where complex lighting interactions with objects are common, leading to frequent occurrences of reflections. Therefore, as suggested in previous work **[1]**, we concentrated our experiments on indoor datasets with reflection-dominant scenes rather than non-reflection-dominant scenes like the KITTI dataset.
>
> >However, we agree that experiments on outdoor scenes, such as KITTI, are important since reflective objects, though uncommon, can still appear in outdoor environments (_e.g.,_ wet road surfaces and reflective building surfaces). Following your suggestion, along with that of Reviewer R3 (tqa3), we have additionally tested our method on widely used datasets such as KITTI (outdoor) and NYU-v2 (indoor), as shown in the table below:
>
> >| Dataset | Method | Abs Rel | Sq Rel | RMSE  | RMSE log | d1   | d2   | d3   |
> >|---|---|---|---|---|---|---|---|---|
> >| KITTI|monodepth2 | 0.118|0.908|4.919| 0.196|**0.871**| 0.957 | 0.980 |
> >| KITTI|monodepth2 + $\cal{L}_{tri}$  | 0.118|0.912|4.943|0.198| 0.867 | 0.956 | 0.980 |
> >| KITTI|monodepth2 + $\cal{L}_{tri}$ + distillation |**0.116**|**0.856**|**4.796**|**0.194**| 0.870 |**0.958**|**0.981**|
> >| NYU-v2|monodepth2  | 0.171 | 0.144  | 0.622 | 0.213 | 0.746 | 0.941 | 0.985 |
> >| NYU-v2|monodepth2 + $\cal{L}_{tri}$ | 0.166   | 0.139  | 0.616 | 0.209    | 0.759 | 0.943 | 0.985|
> >| NYU-v2|monodepth2 + $\cal{L}_{tri}$ + distillation|**0.155**|**0.121**|**0.573**|**0.196**|**0.782**|**0.951**|**0.988**|
>
> >In the case of the KITTI dataset, most scenes exhibit similar attributes and are predominantly composed of non-reflective objects, resulting in limited performance gains in reflective regions. However, as highlighted in our paper, our selective triplet mining loss using reflective region mask $M_r$​ and the distillation process effectively preserved performance in non-reflective areas. Consequently, our method achieved comparable performance to existing approaches with negligible margins. These results are consistent with the trends observed in the ScanNet-NoReflection validation split experiments presented in ```Table 2``` of the main manuscript.
>
> >On the other hand, in the NYU-v2 dataset, where reflections occur more frequently, we observed meaningful performance improvements through our triplet mining loss and distillation method. These results demonstrate that our method is broadly applicable to indoor scenes characterized by complex interactions between light and diverse objects, enabling consistent performance enhancements in such scenarios.
>
> >All experiments conducted on KITTI and NYU-v2 utilized a ResNet-18-based pose network instead of GT pose. This demonstrates that our method operates robustly across both outdoor and indoor scenes, even when incorporating a pose network.
>
> >Additionally, to demonstrate the effectiveness of our method in addressing reflective surface challenges in outdoor scenes, we also evaluate our method on the PittsStereo-RGBNIR dataset **[2]**, an outdoor dataset consisting of challenging materials, including reflective surfaces such as Glass (Reflection), and Glossy (Reflection) types (To see demo images and videos: https://www.cs.cmu.edu/~ILIM/projects/AA/RGBNIRStereo/). For the PittsStereo-RGBNIR dataset, we used RGB images exclusively for testing, as self-supervised monocular depth training was not feasible due to the absence of camera intrinsics and RGB stereo pairs. We performed a zero-shot evaluation on the PittsStereo-RGBNIR dataset using models trained on the KITTI dataset (_i.e.,_ Monodepth2 and Monodepth2 with our methods).
>
> >|Dataset|Method|Avg. (RMSE) | Common|Light| Glass* | Glossy* | Vegetation|Skin| Clothing | Bag |
> >|---|---|---|---|---|---|---|---|---|---|---|
> >|PittsStereo|monodepth2|2.429|2.749|3.505|2.763|3.910|1.549|1.821|1.817|1.318|
> >|PittsStereo|monodepth2 + $\cal{L}_{tri}$|2.359|2.972|3.654|2.573| **3.382**|1.390|1.627|1.745|1.526|
> >|PittsStereo|monodepth2 + $\cal{L}_{tri}$ + distillation|**2.070**|**2.530**|**3.077**|**2.270**|3.493|**1.222**|**1.552**|**1.514**|**0.906**|
>
> >Evaluation of three models trained on the KITTI dataset against the PittsStereo-RGBNIR dataset revealed that our methods achieved significant performance improvement for reflective surfaces, such as **Glass**, and **Glossy** classes. Furthermore, when using our distillation method, we observed substantial performance improvements not only for reflective surfaces but also for non-reflective classes. This validates our claim that the proposed distillation method can substantially enhance the performance on non-reflective surfaces.

---

> ### Author Response · Authors · 2024-11-20
> **Continued response to Reviewer ptjG**
>
> >_**W2, Q2.**_ Proof of assumption: Do reflective regions have low disparities?
>
> >_**A2.**_ Yes, reflective regions consistently exhibit low disparities. According to **[3]**, many stereo and multi-view camera algorithms adhere to a straightforward inverse relationship between 3D depth $Z$ and disparity $d$, expressed as $d = \frac{f \cdot b}{Z}$​, where $f$ is the focal length (measured in pixels) and $b$ is the baseline.
>
> >In reflective regions, the perceived distance from the camera is determined by the combined distance to the actual reflective surface and the distance from the surface to the mirrored object or light source. This cumulative distance is inherently greater than the direct distance to the reflective surface alone (please see ```Figure 1``` in **[1]** for a visual explanation).
>
> >Since distances between distinct objects are always positive, this leads to a larger perceived depth compared to the actual surface distance. Given the inverse relationship between depth and disparity, reflective surfaces consequently exhibit anomalously low disparities relative to the surface itself. This phenomenon leads to a critical issue in SSMDE models trained on reflective regions: they learn incorrect depth estimation, perceiving objects as farther away than the actual surface rather than closer. This systematic error, often referred to as the _“black-hole effect”_, is illustrated in ```Figure 3``` of the main manuscript.
>
> We hope that this explanation sufficiently addresses your concerns and provides clarity on the underlying assumption. We are grateful for the time and insight you've dedicated to reviewing our work, and we look forward to any further feedback that may help us enhance the clarity and rigor of our research.
>
>
> ### **References**
>
> _[1]. Shi, Xuepeng, et al. "3D Distillation: Improving Self-Supervised Monocular Depth Estimation on Reflective Surfaces." Proceedings of the IEEE/CVF International Conference on Computer Vision. 2023._
>
> _[2]. Zhi, Tiancheng, et al. "Deep material-aware cross-spectral stereo matching." Proceedings of the IEEE conference on computer vision and pattern recognition. 2018._
>
> _[3]. Szeliski, Richard. Computer vision: algorithms and applications. Springer Nature, 2022._

---

> > ### Comment · Reviewer_ptjG · 2024-11-25
> >
> > Thank you for addressing my questions on Q2. As for Q1, I believe the results from various experiments effectively demonstrate the advantages of the proposed method.
> >
> > Therefore, I am raising my rating to 6.

---

> > > ### Author Response · Authors · 2024-11-25
> > > **Gratitude for Your Feedback and Evaluation**
> > >
> > > We are very pleased to have been able to adequately address your concerns.
> > >
> > > We believe that the discussions we had with you will greatly contribute to the overall quality of the paper.
> > >
> > > Thank you very much for your careful evaluation and for raising the score.

---

### Author Response · Authors · 2024-11-20
**General response**

We sincerely thank the reviewers for their thorough evaluation of our work and for providing insightful and constructive feedback. We appreciate the time and effort spent reviewing our manuscript, and we value the opportunity to address the concerns and questions raised.

We provide detailed responses to each reviewer’s comment, along with additional experiments and clarifications where necessary. Below, we detail additional experiments conducted per reviewer’s requests:

>In response to **Reviewer ptjG** and **Reviewer tqa3**, we included the experiments that evaluate our method on two common datasets: KITTI and NYUv2. All these experiments were conducted with a pose network instead of GT poses to validate that our method consistently performs well even when involving a pose network, similar to the existing SSMDEs.
>
>Additionally, we have reported experimental results on the pittsStereo-RGBNIR dataset **[6]**, an outdoor scene dataset composed of challenging materials, to analyze the impact of our method on both reflective and non-reflective classes.

>Responding to **Reviewer jMZ7**, we summarized the computational overhead of monodepth2 **[1]** under four scenarios: (1) training with traditional self-supervision, (2) applying our triplet loss, (3) applying our proposed distillation method, and (4) employing the 3D distillation **[2]** approach.

>Addressing the inquiries from **Reviewer jMZ7** and **Reviewer tqa3**, we introduce the experiment results and analysis of the proposed reflective region mask $M_r$. Note that ablation studies of $M_r$ are already included in ```Section D``` of the supplementary material.

>Based on the feedback from **Reviewer tqa3**, we added the experiment results of brand-new methods, including RA-depth **[3]**, GasMono **[4]**, and Lite-mono **[5]**.

We hope that our responses adequately address the reviewers' concerns and further demonstrate the robustness and significance of our contributions. Please do not hesitate to let us know if additional clarifications are needed.

### **References**

_[1] Godard, Clément, et al. "Digging into self-supervised monocular depth estimation." Proceedings of the IEEE/CVF international conference on computer vision. 2019._

_[2] Shi, Xuepeng, et al. "3D Distillation: Improving Self-Supervised Monocular Depth Estimation on Reflective Surfaces." Proceedings of the IEEE/CVF International Conference on Computer Vision. 2023._

_[3] He, Mu, et al. "Ra-depth: Resolution adaptive self-supervised monocular depth estimation." European Conference on Computer Vision. Cham: Springer Nature Switzerland, 2022._

_[4] Zhao, Chaoqiang, et al. "GasMono: Geometry-aided self-supervised monocular depth estimation for indoor scenes." Proceedings of the IEEE/CVF International Conference on Computer Vision. 2023._

_[5] Zhang, Ning, et al. "Lite-mono: A lightweight cnn and transformer architecture for self-supervised monocular depth estimation." Proceedings of the IEEE/CVF Conference on Computer Vision and Pattern Recognition. 2023._

_[6] Zhi, Tiancheng, et al. "Deep material-aware cross-spectral stereo matching." Proceedings of the IEEE conference on computer vision and pattern recognition. 2018._

PittsStereo Dataset Demo: https://www.cs.cmu.edu/~ILIM/projects/AA/RGBNIRStereo/

---

### Author Response · Authors · 2024-11-24
**Friendly Reminder for Discussion Phase Participation**

First, thank you for your constructive reviews on our paper. Your feedback has been invaluable, and we have put our best effort into addressing the points raised in our responses.

We would like to kindly remind you that there are only 3 days left in the authors-reviewers discussion phase for our paper. If you have any issues or additional questions regarding the response, please feel free to ask.

We appreciate your time and effort, and we look forward to your comments.

---

### Meta-Review · Area_Chair_oRHB · 2024-12-21

**Metareview:**

This paper proposes a novel training strategy for an SSMDE, focusing on addressing regions that violate the assumptions of Lambertian reflectance. Initially, the reviewers expressed concerns regarding the need for additional ablation studies and results. However, after the rebuttal, they were satisfied with the additional results provided. All reviewers acknowledged the simplicity and straightforwardness of the method. The AC also reviewed the paper, the feedback, and the rebuttal, and similarly recognized the method as both simple and effective. Therefore, the AC recommends acceptance. The paper would benefit from including additional results and discussions in the final version.

**Additional Comments On Reviewer Discussion:**

All reviewers acknowledged the simplicity and straightforwardness of the method during the rebuttal.

---

### Decision · Program_Chairs · 2025-01-22

Accept (Poster)